# Decrypting the programming of β-methylation in virginiamycin M biosynthesis

Sabrina Collin [1], Russell J. Cox [2], Cédric Paris [3], Christophe Jacob [1], Benjamin Chagot [1] ✉, Kira J. Weissman [1] ✉ & Arnaud Gruez [1] ✉

During biosynthesis by multi-modular *trans*-AT polyketide synthases, polyketide structural space can be expanded by conversion of initially-formed electrophilic β-ketones into β-alkyl groups. These multi-step transformations are catalysed by 3-hydroxy-3-methylgluratryl synthase cassettes of enzymes. While mechanistic aspects of these reactions have been delineated, little information is available concerning how the cassettes select the specific polyketide intermediate(s) to target. Here we use integrative structural biology to identify the basis for substrate choice in module 5 of the virginiamycin M *trans*-AT polyketide synthase. Additionally, we show in vitro that module 7, at minimum, is a potential additional site for β-methylation. Indeed, analysis by HPLC-MS coupled with isotopic labelling and pathway inactivation identifies a metabolite bearing a second β-methyl at the expected position. Collectively, our results demonstrate that several control mechanisms acting in concert underpin β-branching programming. Furthermore, variations in this control – whether natural or by design – open up avenues for diversifying polyketide structures towards high-value derivatives.

The polyketide specialised metabolites of bacteria exhibit a diverse range of biological activities, including antibiotic and anti-cancer properties, and are heavily employed as drugs[1,2]. These highly complex molecules are constructed using an assembly line strategy, in which each task is assigned to a specific enzyme. In the prototypical *cis*-AT systems, the majority of these functions are present within catalytic domains of gigantic multienzymes called polyketide synthases (PKSs)[3]. The functional domains are clustered into modules, where each module is typically responsible for carrying out one round of chain extension and β-processing of the resulting intermediate. In addition to the three domains which are essential to chain building (acyl transferase (AT), ketosynthase (KS), and acyl carrier protein (ACP)), many modules also harbour optional domains which modify the oxidation state of the β-keto group resulting from the condensation reaction. The division-of-labour organisation of PKS systems makes

them attractive targets for synthetic biology approaches aiming at generating high-value derivatives[4].

Relative to the *cis*-AT PKSs, the *trans*-AT systems[5] (Fig. 1) incorporate one or more *trans*-acting enzyme activities and a wider variety of enzymatic functions, including cassettes of enzymes which introduce β-branching into the polyketide intermediates[6]. A common modification is β-methylation, the chemistry of which is reminiscent of the mevalonate pathway of isoprenoid biosynthesis[6,7]. This reaction series involves five discrete proteins (Fig.1): (i) a malonate-loaded ACP (called ACP donor, $ACP_D$); (ii) a condensation-inactive KS domain ($KS^0$) which generates acetyl-$ACP_D$ from the malonyl-ACP; (iii) a 3-hydroxy-3-methylglutaryl-CoA synthase (HMGS) homologue which catalyses attack of the acetate-derived nucleophile on the β-keto group of the polyketide chain attached to an acceptor ACP ($ACP_A$), yielding an HMG-*S*-$ACP_A$ thioester; (iv) an enoyl-CoA hydratase (ECH) homologue

[1]Université de Lorraine, CNRS, IMoPA, F-54000 Nancy, France. [2]OCI & BMWZ, Leibniz Universität Hannover, Schneiderberg 38, 30167 Hannover, Germany. [3]Université de Lorraine, LIBio, F-54000 Nancy, France. ✉e-mail: benjamin.chagot@univ-lorraine.fr; kira.weissman@univ-lorraine.fr; arnaud.gruez@univ-lorraine.fr

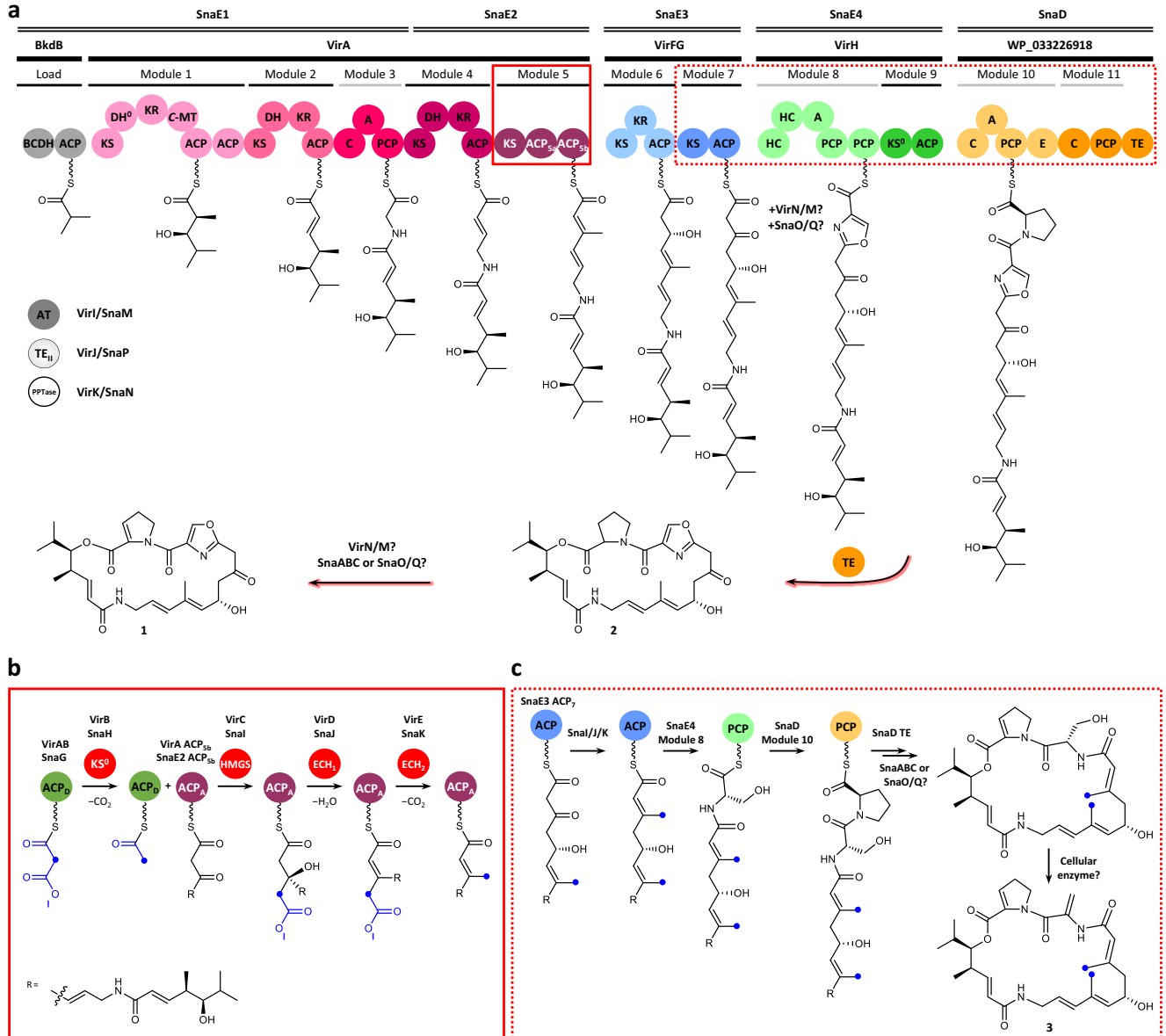

**Fig. 1 | Biosynthesis of metabolites 1–3 in *Streptomyces*. a** Genetic and protein organisation of the biosynthetic pathway to virginiamycin M **1** in *Streptomyces virginiae* (Vir)[15] and *Streptomyces pristinaespiralis* (Sna)[32]. PKS modules are indicated with black bars and NRPS modules with grey bars, and the functional domains within the modules are shown as spheres. Two modules in the system do not catalyse chain extension: module 9 because it incorporates an inactive KS domain (KS[0]); and module 11, as it lacks an A domain for substrate selection. The D-stereochemistry of the Pro is assumed based on the presence of an epimerisation (E) domain in module 10. The enzymes responsible for the dehydrogenation in module 8 to yield the oxazole and the post-assembly line proline dehydrogenation to afford the 2-pyrroline moiety, have not been conclusively identified. Metabolite **2**, the direct precursor of **1**, lacks the proline dehydrogenation. **b** Series of transformations catalysed by the β-methylation cassette acting in module 5 (solid red box), in which acetate generated by decarboxylation of malonate tethered to the ACP donor (ACP_D), is initially condensed with the polyketide intermediate attached to the ACP acceptor (ACP_A) (both the Vir and Sna homologues are indicated). **c** Proposed pathway (dotted red box) leading to doubly methylated (blue dots) derivative **3**, in which a second β-methylation occurs following chain extension in module 7. The proposed structure of **3** is based on its exact mass and the results of feeding isotopically-labelled amino acids (Fig. 7). The configuration of the second β-branch in **3** has been extrapolated from the known stereochemistry of β-branching[7]. Key to the domains/enzymes: BCDH branched-chain α-keto acid dehydrogenase, ACP acyl carrier protein, KS ketosynthase, DH dehydratase, KR ketoreductase, *C*-MT *C*-methyl transferase, C condensation, A adenylation, PCP peptidyl carrier protein, HC heterocyclisation, E epimerisation, TE thioesterase, AT acyl transferase, TE_II proof-reading thioesterase, PPTase phosphopantetheinyl transferase, HMGS 3-hydroxy-3-methylglutaryl-CoA synthase homologue, ECH_1 enoyl-CoA hydratase homologue acting as a dehydratase, ECH_2 ECH acting as a decarboxylase.

(ECH_1) that serves as a dehydratase to produce the corresponding α,β-unsaturated thioester; and finally, (v) a second ECH homologue (ECH_2) that catalyses decarboxylation to afford the β-methyl product. Variation of the electrophile and nucleophile structures, and/or HMG processing sequences, gives access to further types of β-functionality[6,8].

Several intriguing features of β-modification remain to be elucidated. The first concerns how each system selects which

polyketide-ACP_A intermediate to target, as every round of chain extension yields a potential β-keto substrate. In principle, gate-keeping by the HMGS would be sufficient to direct the whole cassette, as no downstream enzymes can act in the absence of this chemistry[9]. Previous work on *trans*-AT PKSs including the mupirocin system identified a sequence motif comprising a conserved Trp flag characteristic of ACP domains in modules targeted for β-methylation[10]. As the majority

of the residues are confined to the domain core, a model was proposed[7,10] in which burial of the Trp side chain governs the orientation of helices α2 and α3 within the ACP four α-helix bundle, allowing both specific residues on helix α3 and the substrate to interact with the HMGS. However, several ACP domains targeted for β-methylation were identified which lacked the conserved Trp (e.g. in the virginiamycin (Vir) M hybrid *trans*-AT PKS-nonribosomal peptide synthetase (NRPS) and the leinamycin PKS)[10], calling into question the proposed recognition mechanism.

A second poorly understood aspect is the presence in typical β-modification modules[7] of repeated (usually two or three) ACP domains. Initial evidence obtained both in vivo and in vitro suggested that they offer a kinetic benefit by allowing for processing in-parallel of multiple intermediates following chain extension[11–13]. However, small-angle X-ray scattering (SAXS) analysis of the β-methylation module within the Vir system[14], revealed that its two ACP domains (ACP$_{5a}$ and ACP$_{5b}$) occupy divergent positions relative to the central, homodimeric KS domain. This architecture suggested that the tandem ACPs might act in-series not in-parallel, with ACP$_{5a}$ participating in the chain extension reaction with the KS, and ACP$_{5b}$ functioning as the way-station during the β-modification reactions.

In this work, we show by Trp fluorescence quenching that the Vir HMGS, ECH$_1$ and ECH$_2$ homologues (VirC, VirD and VirE, respectively) preferentially bind ACP$_{5b}$ in its holo and substrate mimic forms, consistent with the proposed in-series function of the ACPs. Comparative structural analysis of multiple Vir and mupirocin ACPs at high-resolution reveals essentially identical folds, excluding helix α3 orientation as the basis for specific recognition of ACP$_{5b}$. Instead, the crystal structure of the holo-ACP$_{5b}$–VirD complex identifies an ACP interaction motif centred on the phosphopantetheine (Ppant) prosthetic arm and surrounding secondary structural elements, with specificity conferred via distinctive electrostatic surface features of the domain combined with precise Ppant positioning. We also show in vitro that Vir holo-ACP$_7$ is recognised by the cassette enzymes, and identify a doubly β-methylated Vir M derivative in production extracts. The lower titres of the analogue relative to Vir M **1** imply that this second β-methylation is suppressed in vivo. Taken together, our data show that β-methylation programming relies on at least two distinct control modes, but remains imperfect, identifying the deactivation of such mechanisms as a promising strategy for generating polyketide analogues.

## Results

### Binding of VirC, VirD and VirE to Vir ACPs
We first assessed in vitro the interaction between recombinant apo- and holo-ACP$_{5a}$ and ACP$_{5b}$ and VirC, VirD and VirE from the Vir pathway of *Streptomyces virginiae*[15], with ACPs 6 and 7 from non β-methylation modules selected as controls (Supplementary Figs. 1 and 2, Supplementary Data 1 and Supplementary Table 1). While the apo forms are not physiologically relevant, the holo proteins are present at several stages of the catalytic cycles[16]. No binding was detected using tryptophan fluorescence quenching[17] between VirC, VirD and VirE and holo-ACP$_{5a}$, although apo-ACP$_{5a}$ was weakly bound ($K_d$ = 166, 102 and 653 μM, respectively) (Table 1, Supplementary Figs. 3–5). While interaction with apo-ACP$_{5b}$ was similarly weak (77, 178 and 171 μM), the three enzymes showed good affinity towards holo-ACP$_{5b}$ (3.8, 2.9 and 6.8 μM), consistent with an important role in recognition for the Ppant arm.

To provide a more native context to these assays, we also analysed binding of VirC, VirD and VirE to the holo-ACP$_{5a}$-ACP$_{5b}$ didomain (Supplementary Figs. 1 and 2). The observed binding affinities ($K_d$ = 17, 19 and 42 μM) (Table 1, Supplementary Figs. 3–5), are within the same order of magnitude as for the discrete holo domains, and thus we find no evidence for cooperative binding of tandem ACP$_A$ as previously proposed[18]. Furthermore, no binding was detected by VirC, VirD and

**Table 1 | Summary of tryptophan fluorescence quenching results**

| Ligand | Form | VirC $K_d$ (μM) | VirD $K_d$ (μM) | VirE $K_d$ (μM) |
|---|---|---|---|---|
| ACP$_{5a}$ | apo | 166 | 102 | 653 |
| ACP$_{5a}$ | holo | n.d. | n.d. | n.d. |
| ACP$_{5b}$ | apo | 77 | 178 | 171 |
| ACP$_{5b}$ | holo | 3.8 | 2.9 | 6.8 |
| ACP$_{5a}$-ACP$_{5b}$ | holo | 17 | 19 | 42 |
| ACP$_6$ | apo | n.d. | n.d. | n.d. |
| ACP$_6$ | holo | n.d. | n.d. | n.d. |
| ACP$_7$ | apo | 75 | 75 | 301 |
| ACP$_7$ | holo | 18 | 4.3 | 22 |
| ACP$_{5a}$ | acetoacetyl | 40 | – | – |
| ACP$_{5b}$ | acetoacetyl | 4.3 | – | – |
| ACP$_{5a}$ | HMG | – | 51 | – |
| ACP$_{5b}$ | HMG | – | 39 | – |
| ACP$_{5a}$ | methylcrotonyl | – | – | 287 |
| ACP$_{5b}$ | methylcrotonyl | – | – | 30 |
| ACP$_{5a}$ E6761A | apo | 108 | 186 | – |
| ACP$_{5a}$ E6761A | holo | 4.7 | 6.1 | – |
| ACP$_{5a}$ L6764N | apo | 89 | 83 | – |
| ACP$_{5a}$ L6764N | holo | 11 | 4.1 | – |
| ACP$_{5a}$ E6761A/ L6764N | apo | 61 | 117 | – |
| ACP$_{5a}$ E6761A/ L6764N | holo | 10 | 7 | – |
| Sna ACP$_7$ | apo | 78 | 166 | – |
| Sna ACP$_7$ | holo | 13 | 13 | – |

n.d. no binding detected. – = no measurement was performed. The $K_d$ values were determined by fitting the averaged data from two independent measurements (Supplementary Figs. 3–5). HMG (*RS*)-3-hydroxy-3-methyl-glutaryl.

VirE to control ACP$_6$ in either its apo or holo forms. However, unexpectedly, both apo- and holo-ACP$_7$ behaved similarly to the analogous forms of ACP$_{5b}$ with VirC, VirD and VirE (apo-ACP$_7$: 75, 75 and 301 μM; holo-ACP$_7$: 18, 4.3 and 22 μM) (Table 1, Supplementary Figs. 3–5).

Finally, we evaluated binding of VirC, VirD and VirE to ACP$_{5a}$ and ACP$_{5b}$ modified, albeit imperfectly, to mimic the native substrates[19] (Supplementary Fig. 2): acetoacetate (VirC), (*RS*)−3-hydroxy-3-methylglutarate (VirD) and 3-methylcrotonate (VirE). The trends in relative affinities for acetoacetyl-ACP$_{5a/5b}$ and methylcrotonyl-ACP$_{5a/5b}$ were in line with those for the holo proteins (Table 1, Supplementary Figs. 3–5), consistent with the strong preference of the cassette for ACP$_{5b}$, while the presence of substrate analogues did not increase but moderately diminished affinity. Binding by VirD to HMG-ACP$_{5a}$ was likewise weaker than to HMG-ACP$_{5b}$, although the difference was less marked than for the other analogues (51 vs. 39 μM) (Table 1, Supplementary Fig. 4). We excluded an effect on binding to HMG-ACP$_{5b}$ of catalysis, by testing a catalytically-inactive version of VirD (E128Q[20]) (Table 1, Supplementary Fig. 4), which yielded essentially the same $K_d$ (33 μM) as for the wild type protein. Overall, the bulk of the fluorescence quenching data are consistent with preferential binding by the three β-methylation enzymes of ACP$_{5b}$.

### Characterisation of the ACP$_{5b}$/cassette interactions
To gain insight into the determinants of interaction specificity, we aimed to solve the structures of complexes of ACP$_{5b}$ with VirC, VirD and VirE. However, we were unable to obtain crystals with wild type recombinant VirC and VirE, nor with VirC quadruply mutated to promote crystallisation (C114A/Q334A/R335A/R338A) (Supplementary Fig. 1, Supplementary Data 1), as previously described for its

homologue CurD from the curacin pathway[18]. Nonetheless, comparison of small-angle X-ray scattering (SAXS) data obtained on wild type VirC complexed with holo-ACP$_{5b}$, with that calculated[21] from the crystal structure of the acetyl-ACP$_D$–CurD complex (PDB ID: 5KP6)[18], revealed a remarkable fit between the experimental and theoretical scattering curves ($\chi^2 = 1.52$) (Supplementary Fig. 6, Supplementary Table 2). This result shows that the overall structures are similar, implying that HMGS recognition of both ACP$_D$ and ACP$_A$ partners involves common structural elements. In the acetyl-ACP$_D$–CurD case[18], the interface encompasses the entirety of helix α2, the loop α2-α3 and helix α3, as well as a key orientational interaction between the Ppant phosphate and CurD Arg33[10,18].

Next, we successfully solved the structure of VirD alone by Se-SAD at 1.7 Å resolution (PDB ID: 8AHZ) (Fig. 2a, Supplementary Table 3), as well as that of the holo-ACP$_{5b}$–VirD complex at 2.1 Å (PDB ID: 8AHQ) (Fig. 2b). The final VirD model consists of a trimer in the asymmetric unit with r.m.s.d. between monomers of 0.2 Å (202 C$_\alpha$), whose solution relevance was confirmed by SAXS analysis (CRYSOL[21]) (Supplementary Fig. 6, Supplementary Table 2). VirD belongs to the crotonase superfamily whose members exhibit a characteristic fold formed from repeated ββα units[22] (Fig. 2).

In the holo-ACP$_{5b}$–VirD complex (PDB ID: 8AHQ) (Fig. 2b), the asymmetric unit contains two monomers of VirD and two of holo-ACP$_{5b}$. As evidenced by the H3 crystal symmetry, VirD forms characteristic homotrimeric disks[22], two of which are stacked, with six ACPs distributed equatorially at the interface between the trimers. In this arrangement, the smallest gap between S6871 of ACP$_{5b}$ bearing the Ppant (~20 Å) and the catalytic E128 of a VirD monomer is ca. 13.9 Å (Fig. 2c), with the other VirD active sites more than 33.8 Å distant. VirD elements contributing to the interface include the β-strand β10 and the subsequent loop (β10–α4) of one monomer, and the β-turn (β1–β2), the loop (α1–β4) and helix α7 of a second monomer (Fig. 2b, c). Notably, the interface also incorporates the well-folded helix α10 of the first monomer, which is disordered in the structure of VirD alone (Fig. 2a). Concerning the ACP, the interaction involves the C-terminal portion of helix α1, the adjacent loop (α1–α2) and the N-terminal regions of helices α2 and α3. Thus, while complex formation with VirC and VirD involves shared ACP elements (helices α2 and α3), the overall interaction surfaces are distinct. Specific interface residues include T6850 (helix α1), Y6852 (loop α1–α2), D6870, I6872, V6875 and E6876 (helix α2), and Y6895 (helix α3) (Supplementary Fig. 7).

The structure of the complex also identifies key interactions between VirD and the Ppant tethered to S6871 of ACP$_{5b}$ (the distal end of which is not visible in the electron density (Fig. 2c)), consistent with its contribution to binding affinity as observed by fluorescence quenching (Table 1). Specifically, R125 of VirD, whose side chain is oriented by a water molecule, forms a salt bridge with the Ppant phosphate. The same water molecule bridges R192 from an adjacent VirD monomer, which participates in a salt bridge with D6870 of ACP$_{5b}$. The opposite end of the D6870 carboxylate sits in an oxyanion hole comprising the NH groups of I6872 and L6873 of ACP$_{5b}$ helix α2. Overall, these interactions place the oxygen of S6871 within 13.9 Å of the buried catalytic E128 of VirD, and thus within reach of the Ppant arm (Fig. 2c).

Finally, we turned our attention to VirE, studying its interaction with holo-ACP$_{5b}$ in solution by SAXS (Supplementary Table 2) combined with modelling using Colabfold[23] and CORAL[24] (for a full explanation of the analysis, see Supplementary Fig. 6). Briefly, this analysis indicates that VirE exhibits essentially the same overall fold and trimeric structure as VirD, although the C-terminal helix α10 of VirE points towards the solvent instead of covering the active site. This greater flexibility may explain our failure to obtain diffracting crystals of VirE. Furthermore, the obtained SAXS data are consistent with formation of a holo-ACP$_{5b}$–VirE complex ($\chi^2 = 1.89$[25]), in which the ACPs sit at the interfaces between VirE monomers as in the structure of holo-ACP$_{5b}$–VirD. The presence of the complex is further supported by the smaller $R_g$ relative to VirE alone (31.79 vs. 33.01 Å) (Supplementary Table 2), as well as a reduced $D_{max}$ (110.53 vs. 121.90 Å), which are consistent with compaction of VirE upon complex formation. Nonetheless, it remains to study the holo-ACP$_{5b}$/VirE interface at higher resolution in order to precisely elucidate the molecular basis for this interaction.

## Structural basis for ACP$_{5b}$/VirD interaction specificity and ACP anti-selection

Identification of the amino acids in ACP$_{5a}$ corresponding to the ACP$_{5b}$ interface residues shows that with only one exception (V6749 [ACP$_{5a}$] vs. T6850 [ACP$_{5b}$]), they are identical (Supplementary Fig. 7). Thus, this set of residues does not constitute the basis for specific recognition of ACP$_{5b}$ by VirD. We therefore reassessed the hypothesis[10] that ACP recognition might derive, at least in part, from the relative orientation of the α-helices within the domain structures. For this, we solved the NMR structures of holo-ACP$_{5a}$ (PDB ID: 8A7Z), holo-ACP$_6$ (PDB ID: 8AIG) and holo-ACP$_7$ (PDB ID: 8ALL) (Supplementary Table 4), complementing the previously solved apo-ACP$_{5b}$ (PDB ID: 4CA3) and apo-ACP$_{5a}$ (PDB ID: 2MF4) structures[14]. Superimposition of holo-ACP$_{5b}$ (PDB ID: 8AHQ) from the holo-ACP$_{5b}$–VirD crystal structure (Fig. 2b) with apo-ACP$_{5b}$, apo-ACP$_{5a}$, holo-ACP$_{5a}$, holo-ACP$_6$ and holo-ACP$_7$ reveals r.m.s.d. of 0.732 Å (74 Cα), 1.076 Å (59 Cα), 1.095 Å (72 Cα), 1.781 Å (72 Cα), 2.734 Å (72 Cα), respectively. ACPs 5a, 5b, 6 and 7 thus exhibit the same overall organisation including the orientation of the four α-helices (Fig. 3a), an architecture conserved with the previously-characterised Mup ACPs on which the Trp flag model was based[10] (Fig. 3b). Therefore, while the Trp provides strong predictive value for ACP sites of β-branching[10], our results argue against an important role for this residue and the resulting orientation of helix α3, as determinants of cassette interaction specificity with ACP$_A$s. Indeed, both ACP$_{5a}$ and ACP$_{5b}$ contain Phe at this position instead of Trp. This observation is in line with further sequence variability recently uncovered at this position in other *trans*-AT PKS systems (Supplementary Fig. 7)[26–29].

The origin of the observed minor differences in r.m.s.d. lies in the positions of the main chains of the loop regions, particularly α1–α2. This observation prompted us to consider the potential contribution of the α1–α2 loop to recognition. Gratifyingly, close inspection of the holo-ACP$_{5b}$–VirD complex structure (Fig. 2c) identified ACP$_{5b}$ N6865 located in the α1–α2 loop as a potential specificity determinant. The δ-oxygen and nitrogen atoms of N6865 hydrogen bond to two water molecules which are members of a larger, four-molecule water network forming hydrogen bonds to the main chain atoms of ACP$_{5b}$ residues N6865, L6869, D6870, L6873, and L6894. The constraints imposed by this network on the L6869 carbonyl, coupled with those on the D6870 side chain resulting from interaction with R192 of VirD and the ACP helix α2 oxyanion hole, localise the D6870 carboxylate at a distance of 3.9 Å from the phosphate of the Ppant arm. The resulting position adopted by the Ppant to minimise electrostatic repulsion with D6870 apparently favours its efficient interaction with VirD. Notably, in ACP$_{5a}$, polar N6865 is substituted by hydrophobic L6764 (Supplementary Fig. 7), a residue which cannot participate in the hydrogen bond network.

Nonetheless, analysis of the ACP$_6$ and ACP$_7$ sequences reveals that the situation is more complicated than is evident from a single complex structure, as the equivalent sequence position in ACP$_6$ that does not interact with VirD is a Glu, while that in ACP$_7$ which is recognised, is also a Leu (Supplementary Fig. 7). Thus, if a comparable water-mediated hydrogen-bonding network is necessary to establish the correct orientation of the Ppant for binding VirD, other residues in ACP$_7$ can apparently substitute for the Asn of ACP$_{5b}$.

Inspection of the ACP structures also revealed that they diverge in terms of the pattern of charged, hydrophilic and hydrophobic residues on the surfaces adjacent to the Ppant arm (Fig. 4), consistent with

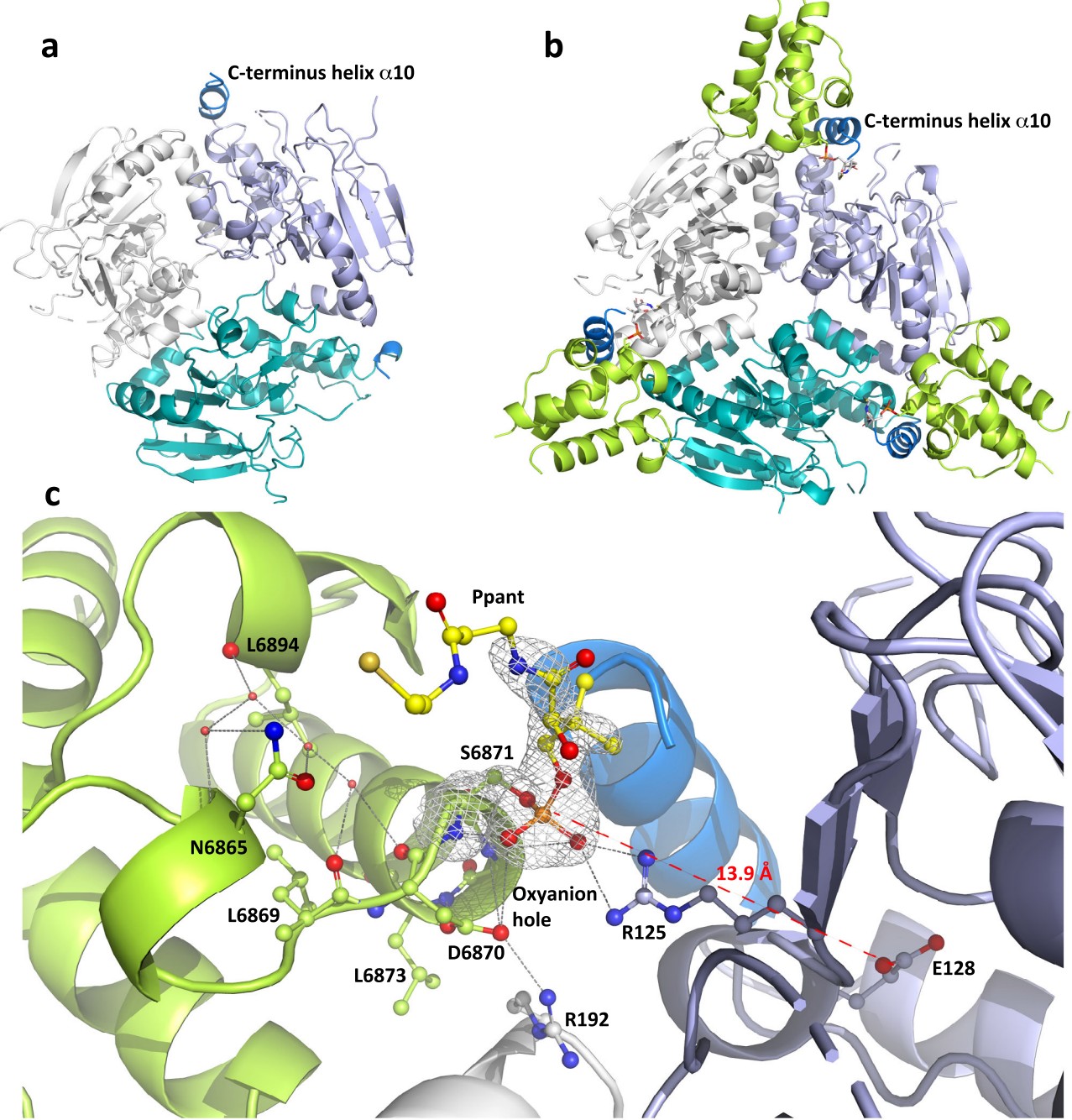

**Fig. 2 | Structural analysis of the VirD/ACP$_{5b}$ interaction, and basis for specificity. a** VirD crystal structure (PDB ID: 8AHZ). The three monomers of VirD are shown in cartoon representation and the three polypeptide chains are coloured in white, teal, and light blue. Helix α10, which is partially defined in the electron density maps, is indicated in marine blue. **b** holo-ACP$_{5b}$−VirD crystal structure (PDB ID: 8AHQ), colour-coded as in (**a**) and with ACP$_{5b}$ shown in lime green. Within the context of the complex, helix α10 is fully structured. The side chains of S6871 and the Ppant arm, only the proximal end of which is visible in the electron density, are shown in stick representation (oxygen atoms are indicated in red, nitrogen atoms in blue, carbon atoms in yellow, and the sulphur atom in gold). **c** Zoom into the VirD active site. The side and main chains of N6865, L6869, L6894 and D6870 are

shown in ball-and-stick representation. The hydrogen bond network between the residues and water molecules (red spheres) is represented as dashed lines. The positively-charged arginines of VirD (R125 and R192) participate in salt bridges with the phosphate moiety and D6870 of the ACP$_{5b}$. The omit map of the Ppant arm and S6871 is contoured at 3σ in white. The distance of 13.9 Å between S6871 and the VirD catalytic E128 is shown as a red dashed line. The oxyanion hole established by the N-terminal portion of helix α3 comprises the NH moieties of D6870, I6872 and L6873. The orientation of the side chain of S6871 is maintained by the oxyanion hole as well as the side chain orientation of D6870. Abbreviation: Ppant phosphopantetheine.

previous observations[18,30]. Notably, in the case of ACP$_{5b}$ (Fig. 4a), the surface surrounding the negatively-charged phosphate group of the Ppant and the adjacent, conserved acidic residue D6870, is largely hydrophobic but punctuated by a protruding hydrophilic region

composed of S6863, N6865 and T6866. This region is itself encircled by three acidic patches, two contributed by the α1−α2 loop (E6854, D6857; D6859, E6861), and the third located at the N-terminus of helix α3 (D6896). ACP$_7$ exhibits an overall similar charge distribution to

**a**

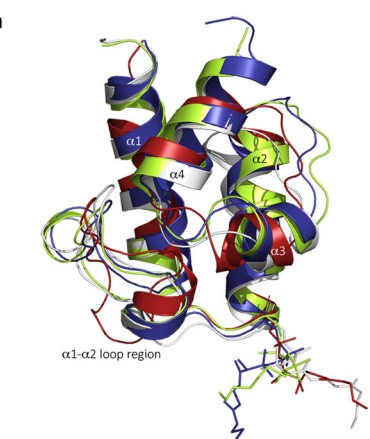

**b**

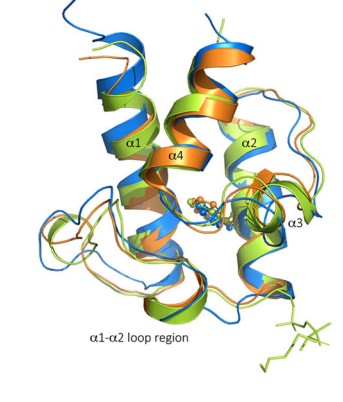

**Fig. 3 | Superimposition of a selection of ACPs on holo-ACP$_{5b}$.**
**a** Superimposition of the average NMR structures of virginiamycin holo-ACP$_{5a}$ (in deep blue), holo-ACP$_6$ (in white) and holo-ACP$_7$ (in firebrick red) on the crystal structure of holo-ACP$_{5b}$ (in lime green). α-helices are shown in cartoon representation and the Ppant cofactor as sticks. **b** Superimposition of the NMR structures of the tandem ACPs of MmpA module 3, Mup ACP$_{3a}$ (in marine blue) and

ACP$_{3b}$ (in orange) (PDB ID: 2L22), on the crystal structure of holo-ACP$_{5b}$ (in lime green) (PDB ID: 8AHQ), reveals r.m.s.d. of 0.949 Å (67 Cα) and 1.106 Å (59 Cα), respectively, and no substantial deviation in terms of the helix α3 orientation. The tryptophan flags of Mup ACP$_{3a}$ and ACP$_{3b}$[10] and the corresponding phenylalanine of Vir ACP$_{5b}$, are shown in ball-and-stick representation.

ACP$_{5b}$ (Fig. 4b). In this case, the hydrophilic patch is replaced by closely co-localised residues R2004, L2007 and E2008, while the ACP$_{5b}$ acidic patch comprising D6859 and E6861 is maintained by ACP$_7$ residues D2001 and D2003. The surface additionally comprises an acidic residue D1995 unique to this domain.

In contrast, in ACP$_{5a}$ (Fig. 4c), the hydrophilic protrusion is less extensive, and flanked by a hydrophobic region comprising F6763 and L6764, while one of the equivalent α1–α2 loop patches contains the positively charged R6756. In addition, residue A6862 in ACP$_{5b}$ is replaced by E6761 in ACP$_{5a}$, contributing an additional negative charge to the surface (Supplementary Fig. 7). Consequently, when the residue at this position is small and hydrophobic it can participate in the ACP core, but when charged, the side chain points towards the solvent. ACP$_6$ differs even more dramatically from ACP$_{5b}$ (Fig. 4d). Specifically, the hydrophilic cluster is replaced by the acidic residue E1218 which is sandwiched between two cationic amino acids, R1206 and R1250, and uniquely among the four ACPs, the domain contains an additional positively-charged residue R1228 near the Ppant phosphate. Thus, both ACP$_{5a}$ and ACP$_6$ exhibit positive net charge in regions which are negatively-charged in ACP$_{5b}$ and ACP$_7$, aggregate electrostatic features which we propose disfavour productive complex formation with the β-cassette enzymes.

## Site-directed mutagenesis supports the specificity model
Taken together, the obtained data suggested a model in which ACP recognition by VirD (and possibly all cassette members) depends principally on the subtle electrostatic landscape of the ACP surface which drives certain interactions, and potentially on the precise positioning of the Ppant arm within the resulting binary complexes, with only a minor role played by the attached substrates. To directly test this idea, we exchanged α1–α2 loop residues E6761 and L6764 of ACP$_{5a}$ with their equivalents in ACP$_{5b}$, A6862 (position contributing to the surface potential) and N6865 (surface hydrophilicity and/or Ppant orientation) (Supplementary Figs. 1 and 2, Supplementary Data 1), and evaluated binding of the single and double mutants to VirD by tryptophan fluorescence quenching. While VirD failed to bind holo-ACP$_{5a}$, it showed good affinity to both of the single holo-ACP$_{5a}$ mutants (E6761A [6.1 μM] and L6764N [4.1 μM]), with $K_d$s comparable to those for binding holo-ACP$_{5b}$ (Table 1). Thus, either single mutation results in VirD recognition. Binding to the double mutant was also observed (7 μM), albeit at slightly reduced affinity, perhaps due to minor perturbation of the ACP$_{5a}$ structure as judged by circular dichroism

(Supplementary Fig. 1). It is also notable that VirD systematically exhibited higher affinity for the holo form of the ACP$_{5a}$ mutants relative to the apo forms (by 4–20-fold) (Table 1), confirming the crucial role of the Ppant cofactor in the interaction.

We also demonstrated that, in contrast to the lack of binding of wild type holo-ACP$_{5a}$ by VirC, both the E6761A and L6764N holo-ACP$_{5a}$ single mutants were recognised (4.7 μM and 11 μM, respectively), while affinity to the double mutant was on par with that of L6764N (10 μM) (Table 1). These data support the idea that the α1–α2 loop region of ACP$_{5b}$ is also critical for its preferential recognition by VirC, although it was not observed previously to lie at the ACP$_D$/CurD interface. While the ACP$_{5b}$–VirC complex evidently resembles that of ACP$_D$–CurD (Supplementary Fig. 6), understanding the detailed role played by these residues in the interaction awaits higher resolution structural information.

## Identification of a doubly β-methylated virginiamycin derivative
The observed binding between ACP$_7$ and VirC, VirD and VirE implied that the ACP$_7$-tethered intermediate may be targeted by the β-methylation cassette in vivo. To evaluate this idea, we scrutinised extracts of a second virginiamycin-producing strain, *Streptomyces pristinaespiralis* ATCC 25486 (Sna cluster, Fig. 1). In contrast to *S. virginiae*, the complete genome sequence of *S. pristinaespiraelis* is available, which is a necessary prerequisite for using CRISPR-Cas9[31] to verify the genetic origin of any detected metabolites, while avoiding off-target effects. To demonstrate the relevance of our interaction studies to this second strain, we measured binding between recombinant (Supplementary Figs. 1 and 2, Supplementary Data 1 and Supplementary Table 1) apo- and holo-Sna ACP$_7$ and VirD. The sequence of VirD shows 63% identity to its Sna homologue, SnaJ[32]. Reassuringly, the $K_d$ determined for the most relevant holo form (13 μM) was essentially identical to that measured for holo-Vir ACP$_7$, while the affinity to the apo-form was twofold weaker (166 (Sna) vs. 75 (Vir) μM) (Table 1).

Next, LC-HRMS analysis of *S. pristinaespiraelis* extracts revealed a signal at $m/z = 526.2912$ ([M + H$^+$]) (rt = 13.56; Fig. 5), in excellent agreement with the calculated for potential analogues of Vir M **1**, incorporating a second β-methyl at C-16 (**3**, Fig. 1). Masses corresponding to alternative doubly β-methylated metabolites were not observed in any significant amounts (Supplementary Fig. 8). Importantly, **3** was no longer detectable in *S. pristinaespiralis* extracts when a portion of the module 7/module 8 interface was deleted using CRISPR-Cas9 (Supplementary Figs. 9 and 10), directly confirming **3** as a

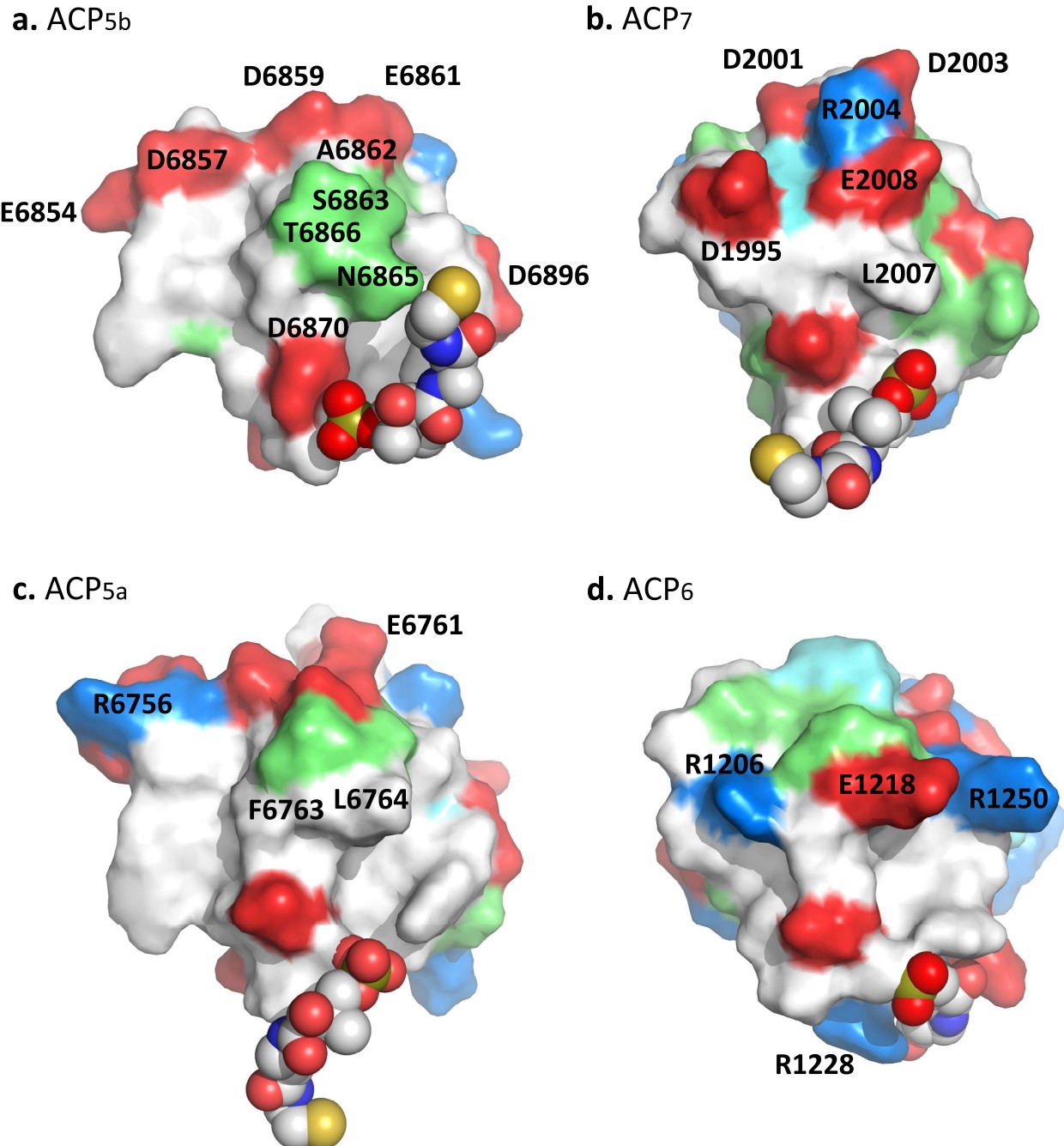

**Fig. 4 | Comparison of the holo-ACP surface features present at the observed interaction interface with VirD.** Surface representation of the holo-ACPs: **a** ACP$_{5b}$ (PDB ID: 8AHQ); **b** ACP$_7$ (PDB ID: 8ALL); **c** ACP$_{5a}$ (PDB ID: 8A7Z); and, **d** ACP$_6$ (PDB ID: 8AIG). The amino acids are coloured according to their properties: positively charged residues (R or K) in marine, histidine in aquamarine, negatively-charged residues (D or E) in red, polar residues (T, S, N) in lime, and hydrophobic residues in white. The Ppant prosthetic group is shown in sphere representation with the oxygen atoms in red, nitrogen atoms in blue, carbon atoms in white, the phosphorus atom in olive and the sulphur atom in yellow.

product of the Sna pathway. Using commercial Vir M as a reasonable calibration standard (Supplementary Fig. 11), we estimated the titres of **3** at 150–200-fold reduced relative to **1** and **2** (Supplementary Table 5). Therefore, while Vir ACP$_7$ is recognised with good affinity by the β-methylation cassette in vitro, the low yield of **3** compared to **1** and **2** shows that this interaction is reduced under native biosynthetic conditions (Fig. 1). Analysis of *S. virginiae* also demonstrated the presence of **3** in addition to **1** and **2**, but at ca. 1000-fold lower yield than **1** from the same strain (Supplementary Fig. 12).

As the low absolute yields of **3** (Supplementary Table 5) precluded purification, to further support its structural assignment, we carried out comparative MS$^2$ analysis of **1**–**3** (Fig. 6, Supplementary Table 5), and fed S. *pristinaespiralis* cultures with isotopically-labelled amino acids, both individually and in combination: L-proline-2,5,5-D$_3$, L-serine-2,3,3-D$_3$, and L-proline-2,5,5-D$_3$ + L-serine-2,3,3-D$_3$. These amino acids were selected to confirm the relatedness of **3** to **1** and **2** (Fig. 1), and simultaneously track the post-incorporation chemistry via loss of deuterium. Comparison of the feeding data obtained on **3** to those of **1**

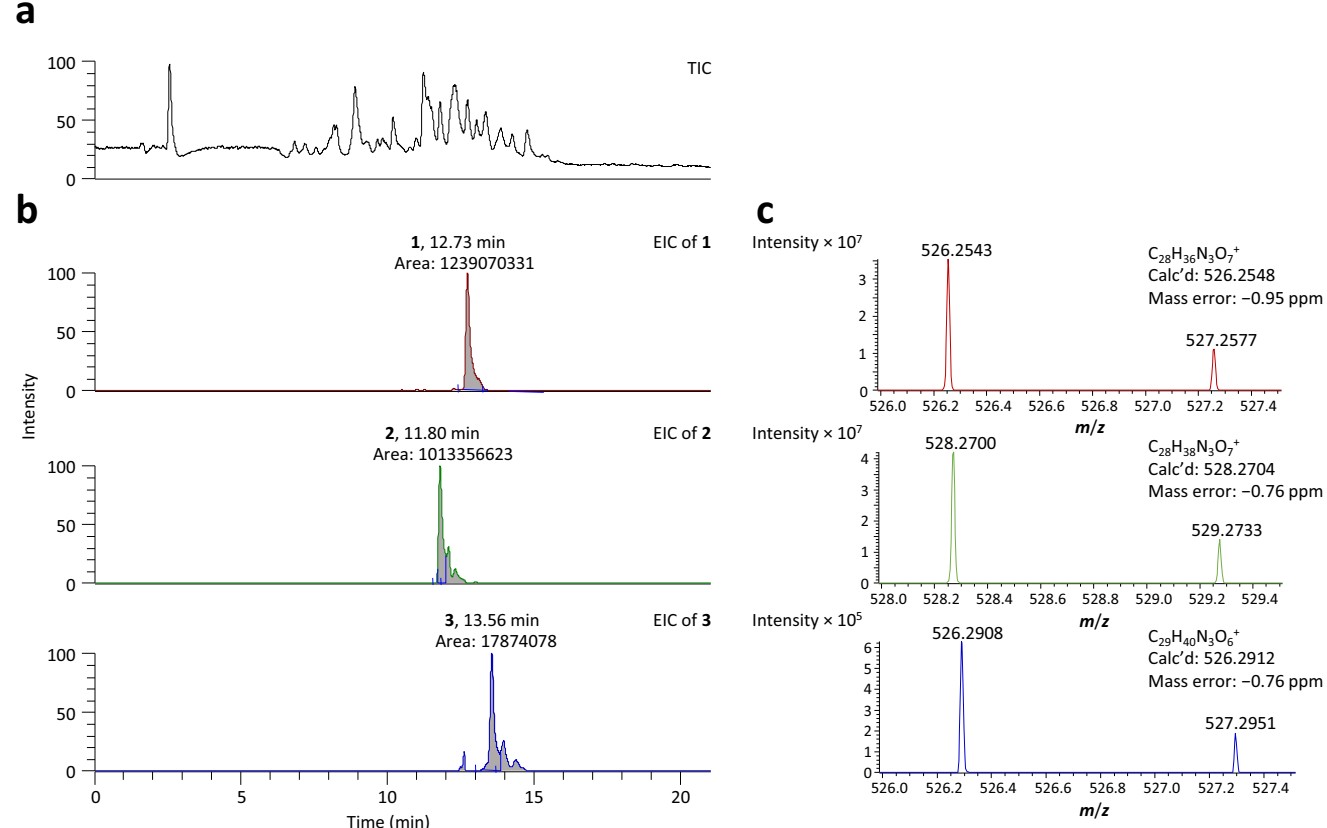

**Fig. 5 | MS analysis of metabolites 1–3 from *S. pristinaespiralis*. a** Total ion chromatogram (TIC). **b** Extracted ion chromatograms (EICs) based on the calculated accurate masses for metabolites **1–3**. In each case, the retention time and integrated peak areas (peaks defined by the vertical blue lines) are indicated. **c** Accurate mass determination of metabolites **1–3**, indicating the calculated and observed masses ($Z=1$), and the mass errors.

and **2** (Fig. 7, Supplementary Fig. 13) demonstrates that **3** incorporates both Ser and Pro residues. Furthermore, the pattern of incorporation into **3** is consistent with retention of two deuteriums from both Ser and Pro. More specifically, the observed Pro labelling provides evidence for incorporation of L-proline-2,5,5-D$_3$ followed by dehydrogenation, as in **1** (Fig. 1). To explain the divergent labelling from Ser, we propose that Ser is incorporated by module 8 as normal, but that the subsequent HC-catalysed heterocyclisation/dehydrogenation does not occur due to mismatched substrate specificity. Indeed, the obtained MS$^2$ data are fully consistent with a structural difference between **1** and **3** in this region (Fig. 6, Supplementary Table 5). Proline is then added by module 10, the product is liberated from the assembly line by macrocyclisation, and the Pro undergoes the native dehydrogenation reaction. Transformation of the Ser to dehydroalanine may be catalysed spontaneously by an adventitious cellular enzyme, explaining the loss of the C-2 proton of the labelled Ser, but retention of the two labels at C-3 (Fig. 7). Indeed, minute quantities of compound potentially corresponding to the non-dehydrated metabolite ($m/z = 544.3017$; rt = 10.49 min) are also observed (Supplementary Fig. 8).

### Evaluation of the relevance of the ACP$_{5b}$/VirD interaction to other PKS systems

As a starting point for our analysis, we reasoned that the specificity determinants governing ACP selection by the β-modification cassette enzymes (e.g. the residues equivalent to ACP$_{5b}$ α1–α2 loop residues A6862 and N6865, and the domain electrostatic surface potential) should be conserved within evolutionarily-linked biosynthetic machineries. In this context, we selected two sets of related systems for which biochemical data were available: the bacillaene/Pks *trans*-AT PKSs[12], and the curacin/jamaicamide *cis*-AT systems[8,13]. In both cases,

the ACPs have been reported in vitro to act in-parallel, which contrasts with the observed in-series function of Vir ACP$_{5a}$ and ACP$_{5b}$. Comparative sequence analysis of the ACPs from these systems provides a ready explanation for this observation, as the ACPs present in the β-methylation modules exhibit high mutual sequence identity (BaeL ACP$_{6a}$ and ACP$_{6b}$: 66%; PksL ACP$_{6a}$ and ACP$_{6b}$: 67%; CurA ACP$_{1a}$ and ACP$_{1b}$: 96%; JamE ACP$_{1a}$, ACP$_{1b}$ and ACP$_{1c}$: 90%; vs. Vir ACP$_{5a}$ and ACP$_{5b}$: 52%). As a consequence, both specificity determinants are well conserved (Supplementary Figs. 14 and 15). It is also notable that the two α1–α2 loop amino acids are rare in the other ACPs of the same PKSs, and never found together (Supplementary Figs. 14 and 15), in agreement with their roles in ensuring specificity. Nonetheless, in the absence of direct study of these systems, we cannot exclude potential interaction between the β-methylation cassette enzymes and additional ACPs located outside of the β-methylation modules, as is the case for Vir ACP$_7$.

We also investigated the ACPs from a large group of *trans*-AT PKS systems related to pederin[5], whose β-methylation modules are believed to share a common evolutionary origin[33]. This analysis revealed different combinations of the two α1–α2 loop amino acids in ACPs nominally targeted for β-methylation (Supplementary Fig. 16 and Supplementary Table 6). While AX (where X = V or I) occurs in four of the nine analysed systems (pederin[5], diaphorin[34], cusperin[35] and nosperin[36]), the remaining modules incorporate RV, VV and VT at the corresponding locations. Comparison of these positions among the tandem or triplet ACPs in these modules further predicts that some of the ACPs act in-parallel (because the critical residues are identical or too similar to allow discrimination by the cassette enzymes), while the remainder function in-series. We also note again that, with only one exception (PedF, ACP$_9$), the pair of amino acids present in the

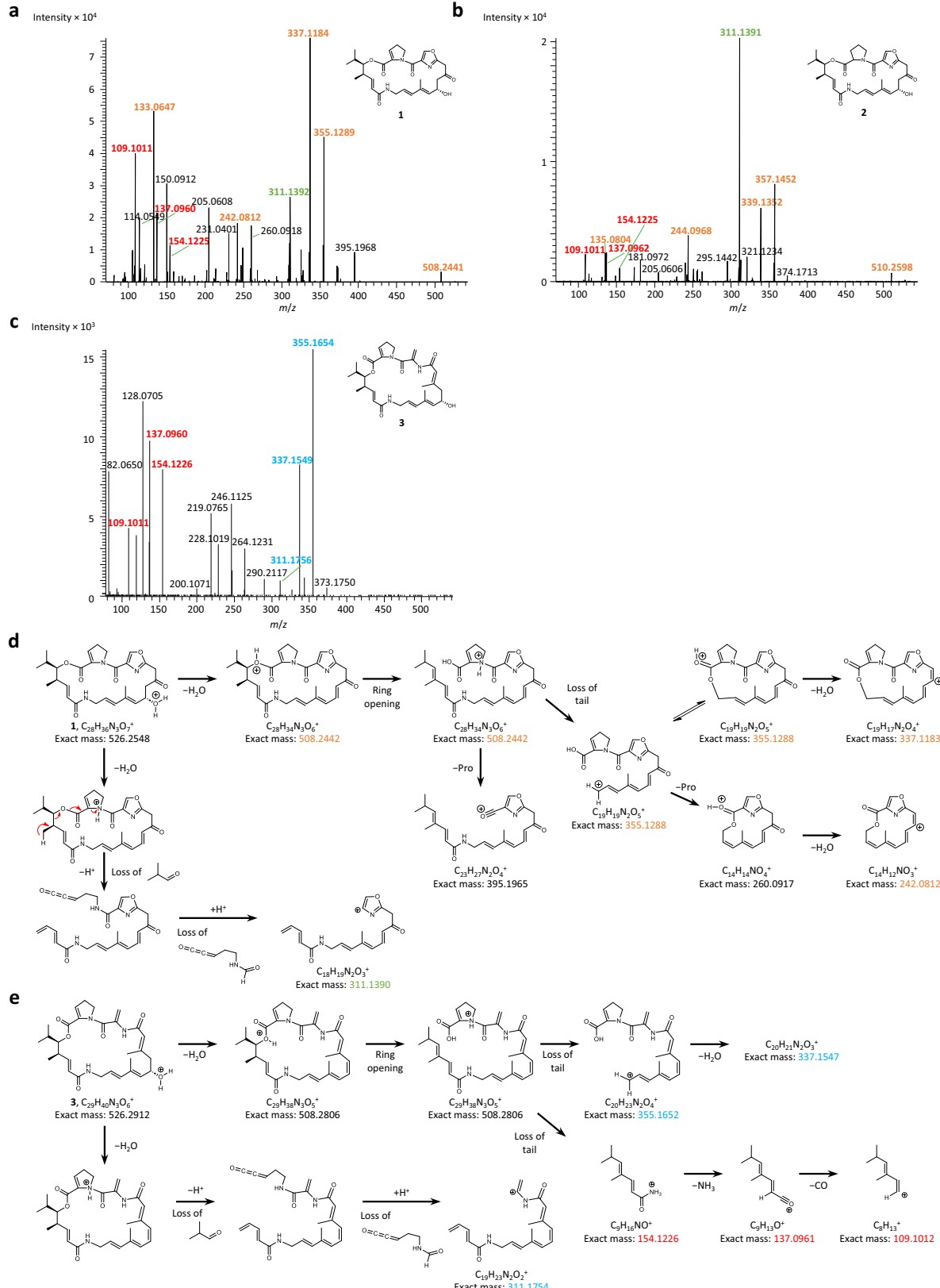

**Fig. 6 | High-resolution MS² analysis of metabolites 1–3.** MS² fragmentation was carried out on the respective parent ions of metabolites: **a 1, b 2** and **c 3**. Fragments differing by 2 Da between **1** and **2**, and therefore likely encompassing the proline/2-pyrroline, respectively, are indicated in orange. Red fragments are common to **1, 2** and **3**, and green to **1** and **2**, while the blue fragments of **3** exhibit the same increase in mass relative to the corresponding fragments of **1** as that seen between the parental molecules **3** and **1**, and therefore must encompass the region which differs between the structures. **d** Proposed fragmentation of **1** ([M + H]⁺) to yield orange and green fragments. **e** Proposed fragmentation of **3** ([M + H]⁺) to yield blue and red fragments.

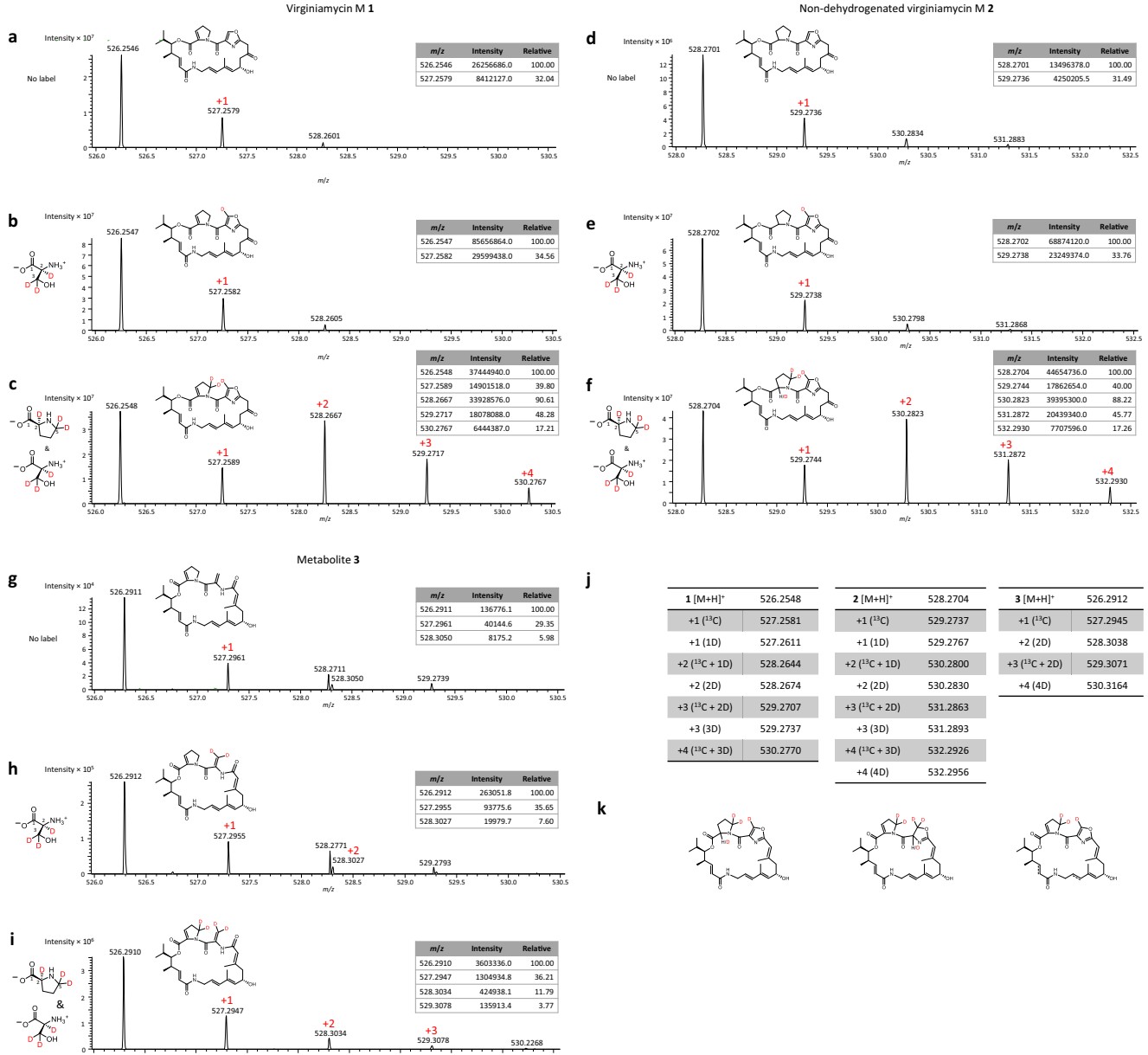

**Fig. 7 | Patterns of isotope incorporation into metabolites 1–3 in the presence of labelled amino acids. a** Mass spectrum of **1** ([M + H⁺]) obtained in the absence of feeding (the calculated masses for **1–3** are presented in Supplementary Table 5). **b** Mass spectrum of **1** ([M + H⁺]) obtained in the presence of fed ʟ-serine-2,3,3-D₃ (the deuterated positions in the Ser and in **1** are shown in red). **c** Mass spectrum of **1** ([M + H⁺]) obtained in the presence of fed ʟ-proline-2,5,5-D₃ and ʟ-serine-2,3,3-D₃. **d** Mass spectrum of **2** ([M + H⁺]) obtained in the absence of feeding. **e** Mass spectrum of **2** ([M + H⁺]) obtained in the presence of fed ʟ-serine-2,3,3-D₃. **f** Mass spectrum of **2** ([M + H⁺]) obtained in the presence of fed ʟ-proline-2,5,5-D₃ and ʟ-serine-2,3,3-D₃. **g** Mass spectrum of **3** ([M + H⁺]) obtained in the absence of feeding. **h** Mass spectrum of **3** ([M + H⁺]) obtained in the presence of fed ʟ-serine-2,3,3-D₃. **i.** Mass spectrum of **3** ([M + H⁺]) obtained in the presence of fed ʟ-proline-2,5,5-D₃ and ʟ-serine-2,3,3-D₃. In panels **a**–**i**, the relative peak intensities are shown in inset. At the resolution at which these experiments were carried out (60 K, 200 *m/z*, full width at half maximum (FWHM), which translates into a resolution of 40 K at 500 *m/z*), it was not possible to distinguish between two ions which have a Δ*m/z* = 0.0030 AMU, which is the case with metabolites differing by the alternative presence of D (+1.0063) and ¹³C (+1.0033). At 40 K, discriminating between such ions would have required a mass difference of 0.0125 AMU or greater (i.e. 4× the

theoretical difference between D and ¹³C). To reflect this ambiguity, the peaks in the spectra have been labelled +1, +2, etc. to indicate that they potentially represent a mixture of isotopically labelled species arising from natural abundance ¹³C and incorporation of deuterium (the respective calculated masses are shown in panel **j**). Nonetheless, the presence of the labelled amino acids is clearly demonstrated by: i) the increase in intensity of the +1 peaks in the presence of ʟ-serine-2,3,3-D₃, and ii) the appearance of +2, +3 (and +4 peaks in the case of **1** and **2**) when both ʟ-serine-2,3,3-D₃ and ʟ-proline-2,5,5-D₃ are fed. The relatively low levels of amino acid incorporation into **3** coupled with weak signal intensity, impeded observation of simultaneous incorporation of ʟ-serine-2,3,3-D₃ and ʟ-proline-2,5,5-D₃ (peak at +4). Globally, the distinct isotopic patterns observed for **3** relative to **1** and **2** are consistent with incorporation of two deuteriums from both Pro and Ser, and therefore with the proposed divergence in post-incorporation chemistry occurring on the Ser. **k** Alternative structures for **3** with identical exact masses ([M + H⁺] = 526.2912), and which can be explained using classical PKS/NRPS biochemistry. In each case, the labelling patterns (red) expected in the presence of fed combined ʟ-proline-2,5,5-D₃ and ʟ-serine-2,3,3-D₃ are indicated. The third structure includes a single site of saturation as indicated by the dashed lines, which could be introduced spontaneously by a cellular reductase.

β-methylation ACPs is never found within the other ACPs of the same PKSs (Supplementary Fig. 16 and Supplementary Table 6), adding weight to the idea that they mediate specificity.

In terms of the associated VirD homologues from these systems, we considered the amino acids equivalent to VirD residues R125 and R192 which are involved in Ppant positioning, as well as R238 which forms a critical salt bridge with $ACP_{5b}$ E8676 located on helix α3 (Fig. 2c) (for simplicity, all residues in the homologues will be referred to using the VirD numbering). Concerning BaeH and PksH, this analysis notably revealed that R238 is conserved. Unexpectedly, neither R125 nor R192 is present in BaeH and PksH (Fig. 2c). However, inspection of these sequences reveals residue substitutions which notably lie close to the phosphate moiety in the holo-$ACP_{5b}$–VirD complex (Fig. 2), that could offset the missing positive charges. In the case of R125, these include R244 in BaeH, and K159 in both BaeH and PksH, while R192 may be compensated for by R191 present in both sequences (Supplementary Fig. 17). A similar situation is evident with the pederin family of VirD homologues (Supplementary Fig. 18), with a high proportion of sequences containing R238 (or a directly upstream positively-charged residue), and R192 or R191. While none of these homologues includes R125, compensating residues can be identified at either position 159 or 244 in multiple cases. The one exception concerns the onnamide $ECH_1$ homologue, which is not present as a discrete enzyme, but as a domain within the subunit OnnB[37]. In this case, the $ECH_1$ acts in *cis* and therefore does not need to distinguish between the β-methylation ACPs and those in other modules – an observation which potentially explains the complete lack of conservation of the three Arg residues. In summary, these data argue that the $ECH_1$ elements mediating interaction with $ACP_A$ domains in *trans* are conserved among *trans*-AT PKS systems, as variation in these residues is compensated for by charged amino acids present elsewhere in the structures.

This analysis can be extrapolated to some extent to the curacin and jamaicamide VirD homologues, CurE and JamI, respectively (Supplementary Fig. 19)[8,13]. Residue R238 is present in JamI, and may be substituted for by K237 in CurE, while both enzymes contain R191 instead of R192. The one substantial divergence concerns position 125, which is V in both homologues, and for which no clear compensating amino acid can be identified at either position 159 or 244. Nonetheless, we note that helix α10 and the downstream residues of both enzymes, which are among the most divergent regions of the proteins relative to VirD (Supplementary Fig. 19), contain multiple Lys residues. These positively-charged amino acids could conceivably replace R125 if helix α10 were positioned alternatively. Despite this potential difference in the mode of VirD recognition, our analysis suggests that at least several $ECH_1$/$ACP_A$ interaction motifs are common to both *cis*-AT and *trans*-AT PKSs.

## Discussion

Diverse β-modification reactions occur during biosynthesis by many *trans*-AT PKSs, and certain *cis*-AT systems[5,13,38]. How specificity is achieved is an intriguing question, as the β-modification cassette enzymes must distinguish between a large number of potential acceptor ACP ($ACP_A$) domains bearing β-keto substrates. A further layer of complexity is the typical presence in β-branching modules of 2–3 $ACP_A$s, implying that one or all of these domains could serve as the site for the reaction series[11]. Understanding how acyl-ACP substrates are chosen, or conversely counter-selected, is a prerequisite to introducing β-modification reactions at specific alternative positions in polyketides by genetic engineering.

In this work, we investigated the β-methylation module 5 present in the virginiamycin (Vir) *trans*-AT PKS-NRPS, which comprises a KS domain and tandem ACPs ($ACP_{5a}$ and $ACP_{5b}$) (Fig. 1). Both ACP domains contain a Trp to Phe substitution at a residue position previously proposed to be critical for flagging the *trans*-AT PKS $ACP_A$s at which β-modification should occur[10] (Supplementary Fig. 7), raising

the questions of how they are recognised by the Vir cassette enzymes. Furthermore, the higher accessibility of $ACP_{5b}$ as revealed by the module 5 SAXS structure[14], suggested that it might be the preferred site of β-modification in *trans*.

We show here that β-methylation cassette members VirC, VirD and VirE do indeed preferentially recognise $ACP_{5b}$, even when the ACP is excised from its modular context, and that β-modification occurs within defined $ACP_{5b}$/partner complexes (Fig. 2, Supplementary Fig. 6). The fact that VirD and VirE prefer $ACP_{5b}$ also demonstrates that the gate-keeping function within the cassette is not limited to the HMGS VirC. The crystal structure of the holo-$ACP_{5b}$–VirD complex (Fig. 2b, c) in combination with comparative sequence analysis (Supplementary Fig. 7), further reveals that the key $ACP_{5b}$ interface residues are highly conserved with $ACP_{5a}$. $ACP_{5b}$ selectivity instead derives in large measure from the electrostatic character of the surrounding amino acids which drive complex formation[39] (Fig. 4). Hydrogen-bonding restraints imposed on the Ppant cofactor (Fig. 2c) may additionally optimise ACP/cassette interactions, but the attached substrates appear to contribute only minimally to the binding affinity[18] (Table 1).

Overall, this specificity for $ACP_{5b}$ likely ensures that two ACPs act principally in-series to support, respectively, chain extension and β-modification. This mechanism would require that the β-keto intermediate be transacylated between the two ACPs, a transfer that is compatible with the measured inter-ACP distance[14]. It would also necessitate that the holo form of $ACP_{5b}$ be present, but there is precedent for this in *trans*-AT PKS systems[40]. While we can only speculate as to possible explanations, holo-$ACP_{5b}$ may not be an efficient substrate for malonylation by the *trans*-acting AT (VirI/SnaM), and/or it may be poorly accessible to the AT due to preassembly of complexes between $ACP_{5b}$ and the β-methylation cassette enzymes.

In contrast to the in-series behaviour of Vir $ACP_{5a}$ and $ACP_{5b}$, previous work has provided evidence for the in-parallel action of certain β-methylation ACPs[8,11–13]. In the case of the Bae/Pks and Cur/Jam systems, this observation is not surprising, as the multiple ACPs present in the modules targeted for β-methylation are highly similar. The large pederin family of *trans*-AT PKSs offers a more interesting test case, because the key α1–α2 loop residue positions are not well conserved. This observation implies that a subset of these ACPs function in-parallel and the remainder in-series. Similarly, the VirD homologues within this family share the main $ACP_A$ interaction elements, but the precise positions of the involved residues differ among the systems. Globally, the presence of multiple solutions to the ACP/β-methylation cassette recognition problem in the pederin family systems despite their presumed common evolutionary origin, is consistent with the earlier proposal[33] that the parental gene cluster diverged substantially following extensive inter-phylum transfer.

Given the prevailing view in the literature that β-modification occurs with high fidelity[7], we were surprised to observe that $ACP_7$ is also efficiently recognised by the three cassette enzymes in vitro, an interaction which translates in vivo in two strains of *Streptomcyes* into a Vir M analogue **3** bearing a second β-methyl group. Notably, titres of **3** at ca. 0.1–1% of those of **1** (Supplementary Fig. 12, Supplementary Table 5), are on par with amounts of polyketides typically obtained by PKS genetic engineering[4]. While this result might be interpreted as indicating that the pathway is intrinsically diversity-oriented, the fact that **3** titres are lower than those of **1** and **2** rather argues that **3** arises from intermittent failures to suppress $ACP_7$/cassette interactions. This phenomenon likely exemplifies the evolutionary challenges of achieving catalytic fidelity with acyl-ACP substrates which must interact with multiple partners, given the limited number of secondary structure and surface features offered by the small (ca. 10 kDa) domains[39]. This problem may be further aggravated by the evident structural plasticity of VirD (Fig. 2a, b) and VirE which derives from helix α10, which could allow them to adapt to alternative partners. Furthermore, module 7, which comprises only KS and ACP domains, is

the sole PKS module in the Vir/Sna systems not to incorporate *cis*-acting modification domains which could kinetically outcompete β-methylation (Fig. 1). Nonetheless, control of β-methylation in *S. virginiae* is evidently tighter than in *S. pristinaespiralis*, as *S. virginiae* produces proportionally lower amounts of **3**.

In this context, we hypothesise that in addition to preferential recognition of ACP$_{5b}$, the atypical[5] domain composition of module 8 also plays a role in β-methylation programming. Notably, this module incorporates two copies of precisely the domains—heterocyclisation (HC) and peptidyl carrier protein (PCP) (Fig. 1)—required for extension of the module 7 intermediate followed by oxazoline formation, suggesting that these domains kinetically and/or sterically outcompete the cassette enzymes, albeit imperfectly. It may be noteworthy that kinetic arguments are now also used to explain complex programming in iterative PKSs[41] and NRPS systems with *trans*-acting components[42]. Given that a high proportion of *trans*-AT PKSs systems comprise *trans*-acting enzymes including but not limited to β-branching cassettes[5,7], it is likely that the existence of multiple control mechanisms is not limited to the virginiamycin system. We thus propose deblocking these latent chemistries as an innovative strategy for further diversifying polyketide structures.

## Methods

### Bioinformatics analysis
*trans*-AT PKSs containing β-methylation modules were identified using refs. [5,7]. For comparative analysis of ACP domains, all PKS subunit sequences (with the exception of VirFG[14]) were retrieved from the Protein data base (http://www.ncbi.nlm.nih.gov/protein), and domain boundaries were established relative to the solved structures of Vir ACPs 5a and 5b (PDB IDs: 2MF4, 4CA3)[14]. Sequence alignments shown in figures were generated using the NPS@ web server (https://npsa-prabi.ibcp.fr/cgi-bin/npsa_automat.pl?page=/NPSA/npsa_clustalw.html)[43] and the alignment figures created with ESPript[44].

### Materials and DNA manipulation
Biochemicals and media were purchased from VWR (glycerol, NaPi, NaCl, MgSO$_4$), BD (tryptone, yeast extract), Thermo Fischer Scientific (Tris, EDTA), Euromedex (isopropyl β-D-1-thiogalactopyranoside (IPTG)), and Sigma-Aldrich (betaine, imidazole, Tris(2-carboxyethyl) phosphine hydrochloride (TCEP), starch), and Roquette (corn steep). L-proline-2,5,5-D$_3$ and L-serine-2,3,3-D$_3$ were sourced from CDN Isotopes. The enzymes for genetic manipulation were purchased from Thermo Fisher Scientific. Isolation of DNA fragments from agarose gel, purification of PCR products and extraction of plasmids were carried out using the NucleoSpin® Gel and PCR Clean-up or NucleoSpin® Plasmid DNA kits (Macherey Nagel). Standard PCR reactions were performed with Phusion High-Fidelity DNA polymerase (Thermo Fisher Scientific); and reactions were carried out on a Mastercycler Pro (Eppendorf). DNA sequencing was carried out by Eurofins.

### Strains and media
*Escherichia coli* BL21(DE3) strains (Supplementary Table 1) were obtained from Novagen and were cultured in LB medium (yeast extract 10 g L$^{-1}$, tryptone 5 g L$^{-1}$, NaCl 10 g L$^{-1}$, adjusted to pH 7.0 with NaOH) or on LB agar plates (LB medium supplemented with 20 g L$^{-1}$ agar) at 37 °C. *Streptomyces pristinaespiralis* ATCC 25486 (DMSZ, Germany) and the derived mutants were sporulated on RP agar plates (20 g L$^{-1}$ starch, 20 g L$^{-1}$ soybean flour, 0.5 g L$^{-1}$ valine, 0.5 g L$^{-1}$ K$_2$HPO$_4$, 1 g L$^{-1}$ MgSO$_4$ × 7H$_2$O, 2 g L$^{-1}$ NaCl, 3 g L$^{-1}$ CaCO$_3$, 20 g L$^{-1}$ agar in tap water) for 7 days at 30 °C. All strains were maintained in 20% (v/v) glycerol and stored at −80 °C. *E. coli* ET12567/pUZ8002 was used for conjugation and appropriate antibiotics were added to LB liquid and agar cultures at the following concentrations: ampicillin 100 mg L$^{-1}$, kanamycin 50 mg L$^{-1}$, apramycin 25 mg L$^{-1}$, chloramphenicol 25 mg L$^{-1}$ and nalidixic acid 25 mg L$^{-1}$. For metabolite production by *S. pristinaespiralis*

and its mutant, and *S. virginiae* MAFF No. 116014 (Genebank Project, National Institute of Agrobiological Sciences, Japan) (Supplementary Table 1), 20 μL of spores (or 1 mL of mycelium in the case of *S. virginiae*) were used to inoculate 25 mL innoculation medium (10 g L$^{-1}$ corn steep powder, 15 g L$^{-1}$ saccharose, 10 g L$^{-1}$ (NH$_4$)$_2$SO$_4$, 1 g L$^{-1}$ K$_2$HPO$_4$, 3 g L$^{-1}$ NaCl, 0.2 g L$^{-1}$ MgSO$_4$ × 7H$_2$O, 1.25 g L$^{-1}$ CaCO$_3$ in tap water, pH 6.9), followed by incubation at 30 °C and 180 rpm on rotary shaker for 72 h. Production medium (25 g L$^{-1}$ soybean flour, 7.5 g L$^{-1}$ starch, 22.5 g L$^{-1}$ glucose, 3.5 g L$^{-1}$ yeast extract, 0.5 g L$^{-1}$ ZnSO$_4$ × 7H$_2$O, 6 g L$^{-1}$ CaCO$_3$ in tap water, pH 6.0) was inoculated with 2% of precultures, and incubated at 30 °C, 180 rpm on a rotary shaker for 96 h. To evaluate its effect, certain cultures were supplemented with 2% XAD-16 resin (Sigma-Aldrich). For feeding experiments, cultures were supplemented individually with L-proline-2,5,5-D$_3$ or L-serine-2,3,3-D$_3$, or a combination of L-proline-2,5,5-D$_3$ and L-serine-2,3,3-D$_3$, at 4, 24 and 48 h after incubation, in equal portions, to a final concentration of 3 mM.

### Gene cloning and site-directed mutagenesis
All protein-encoding constructs were amplified directly from *Streptomyces virginiae* genomic DNA using forward and reverse primers incorporating *Bam*HI and *Hind*III restriction sites, respectively (Supplementary Data 1), and were ligated into the corresponding sites of vector pBG-102 (with the exception of VirC and its quadruple mutant which were cloned into pLM-302). Vector pBG-102 codes for a His$_6$-SUMO tag and pLM-302 codes for a His$_6$-maltose binding protein (MBP) tag (Centre for Structural Biology, Vanderbilt University). In both cases, cleavage of the tags resulted in a non-native N-terminal Gly-Pro-Gly-Ser sequence. The sequences of all constructs were verified by DNA sequencing prior to protein expression studies. Site-directed mutations were introduced into ACP$_{5a}$ and VirD by PCR using mutagenic oligonucleotides (Supplementary Data 1) and Phusion High-Fidelity polymerase, followed by digestion of the parental DNA by 1 μL of *Dpn*I Fast digest (Thermo Fischer Scientific). The presence of the correct mutations was confirmed by sequencing.

### Expression and purification of recombinant proteins ACP domains, VirC, VirC quadruple mutant (C114A/Q334A/R335A/R338A), VirD, VirD E128Q and VirE
All constructs were transformed into *E. coli* BL21(DE3) cells and grown at 37 °C in LB medium supplemented with 50 μg mL$^{-1}$ kanamycin to an A$_{600}$ of 0.8, and then IPTG added to a final concentration of 0.5 mM. Following incubation at 20 °C for 18 h, the cells were harvested by centrifugation at 3000 × *g* for 30 min at 4 °C, and cell pellets stored immediately at −80 °C. Vir ACP$_{5a}$ and ACP$_{5b}$ and all APC$_{5a}$ mutants, ACP$_{5a}$–ACP$_{5b}$ didomain, ACP$_6$ and ACP$_7$ were purified using the same method[14]. Specifically, cells were resuspended in buffer 1 (50 mM sodium phosphate (pH 7.5), 250 mM NaCl), lysed by sonication, and cell debris were removed by centrifugation and filtration (0.45 μm). The cell lysates were then loaded onto a HisTrap 5 mL column (GE), which had previously been equilibrated in buffer 1. The column was washed extensively with buffer 1 containing 75 mM imidazole, and the His-tagged proteins were eluted at 350 mM imidazole. Incubation was then carried out with His-tagged human rhinovirus 3 C protease (1 mM) for 12–16 h at 4 °C in order to cleave the affinity-solubility tags. The target constructs were then separated from the remaining His-tagged proteins via loading onto a HisTrap 5 mL column (GE), followed by elution in buffer 1 containing 20 mM imidazole. Final polishing was carried out by size-exclusion chromatography using a Superdex 75 26/60 column (GE) in buffer 1.

In the case of all proteins of the β-methylation cassette, the cell pellets were resuspended in buffer 2 (50 mM NaPi pH 7.5, 250 mM NaCl, 10% glycerol for VirC and the VirC quadruple mutant, or 20 mM Tris-HCl pH 8.5, 300 mM NaCl, 10% glycerol (VirD, VirD E128Q and VirE)) containing 8 U mL$^{-1}$ of Benzonase (Merck) and 5 mM MgSO$_4$. Cells were lysed by sonication and clarified by centrifugation

(35,000 × g for 40 min). Cell extracts were loaded onto a 5 ml HisTrap column (Cytiva) and washed with buffer 2 supplemented with 20 mM imidazole. The supernatant was loaded onto a HisTrap 5 mL column equilibrated with buffer 2 using an Akta Pure system (Cytiva). The proteins were eluted using a linear gradient of 0–50% buffer 3 (50 mM NaPi pH 7.5, 250 mM NaCl, 300 mM imidazole for VirC and the VirC quadruple mutant or 20 mM Tris-HCl pH 8.5, 300 mM NaCl, 300 mM imidazole (VirD, VirD E128Q and VirE)) over ten column volumes.

All of the His₆-tagged constructs were then incubated with His-tagged human rhinovirus 3 C protease (1 μM) for 12–16 h at 4 °C to cleave the affinity/solubility tags. The constructs were then separated from the remaining His-tagged proteins by loading onto a HisTrap 5 mL column, followed by elution in buffer 2 containing 20 mM imidazole. VirD, VirD E128Q and VirE were subsequently injected onto a Q-sepharose column (trimethylammonium on 6% agarose) equilibrated in buffer (20 mM Tris-HCl pH 8.5, 20 mM NaCl, 10% glycerol). All proteins were then eluted using an NaCl gradient (100 mM to 1 M) at 5 mL min⁻¹. Eluted fractions found to contain protein of the correct molecular weight as judged by SDS-PAGE analysis were pooled, concentrated using an Amicon Ultracel-10 (Merck Millipore) by centrifugation at 4000 × g, and loaded onto either a Superdex 200 16/60 (Cytiva) (VirD, VirD E128Q and VirE) or a Superdex 75 16/60 column (Cytiva) (VirC and the VirC quadruple mutant), equilibrated with 20 mM Tris-HCl pH 8.5, 300 mM NaCl, 5% glycerol. Following a concentration step, the purity of the recombinant proteins was determined by SDS-PAGE (Supplementary Fig. 1), and their concentrations were determined by NanoDrop (or Qubit for ACP₆) (Thermo Scientific), with extinction coefficients calculated using the ExPASy ProtParam tool[45].

### Expression of labelled protein samples for structural biology
Seleniated wild type VirD was produced in M9 minimal medium (50 mM Na₂HPO₄, 22 mM KH₂PO₄, 10 mM NaCl, 20 mM NH₄Cl, adjusted to pH 7.2 with NaOH) for SAD/MAD phasing. Autoclaved M9 medium was supplemented with 50 mg L⁻¹ of thiamine and riboflavin, 4 g L⁻¹ glucose, 100 μM CaCl₂, 2 mM MgSO₄, 40 mg L⁻¹ selenomethionine, and 40 mg L⁻¹ of the 19 amino acids, based on the methionine biosynthesis inhibition method[46]. ¹³C,¹⁵N-enriched Vir ACP₅ₐ, ACP₆ and ACP₇ were produced in M9 medium supplemented with ¹⁵NH₄Cl (0.5 g L⁻¹) and ¹³C-glucose (2.0 g L⁻¹), as the only sources of nitrogen and carbon. The labelled proteins were purified to homogeneity as described above.

### Svp-catalysed modification of ACP domains and verification by HPLC-MS
Following size-exclusion chromatography, apo-ACPs (1 mM) were incubated in buffer (20 mM Tris-HCl pH 8.5) with 5 mM (acyl-)CoASH, 40 μM PPTase Svp[19], 10 mM MgCl₂ and 50 mM TCEP for 22 h at 20 °C. The ACPs were then purified using a Superdex 75 16/60 column (Cytiva) equilibrated in 20 mM Tris-HCl pH 8.5, 250 mM NaCl, 50 mM TCEP. Quantitative modification was verified for all of the ACPs by HPLC-MS (Supplementary Fig. 2) using either a Thermo Scientific Orbitrap ID-X Tribrid Mass Spectrometer, or an LTQXL mass spectrometer, both equipped with an in-line photodiode array detector (PDA) and an atmospheric pressure ionisation interface operating in electrospray mode (ESI). All samples were diluted with Milli-Q water to a concentration of 50 μM and injected onto an Alltima™ C18 column (2.1 × 150 mm, 5 μm particle size). Analysis was carried out with Milli-Q water containing 0.1% TFA (A) and acetonitrile containing 0.1% TFA (B), using the elution profile: 0–15 min, linear gradient from 10–98% solvent B; 15–20 min, constant 98% solvent B; 20.1–26 min, constant 10% solvent B. In the case of the LTQXL, MS scans were performed in ESI⁺ in the mass range m/z = 100–2000, at 3 K resolution, with MS parameters as follows: spray voltage, 5 kV; source gases were set respectively for sheath gas, auxiliary gas and sweep gas to 20, 5 and 5 arbitrary units

min⁻¹; capillary temperature, 350 °C; capillary voltage, 7 V; tube lens, split lens and front lens voltages 180 V, −22 V and −11.75 V, respectively. MS data acquisition was carried out using the Xcalibur v. 2.1 software (Thermo Scientific). For the Orbitrap, MS scans were performed in heated ESI positive ion mode (HESI⁺) in the mass range m/z = 150–2000, at 7.5 K or 60 K resolution (full width of the peak at its half maximum, fwhm, at m/z = 200) with MS parameters as follows: spray voltage, 4 kV; source gases were set respectively for sheath gas, auxiliary gas and sweep gas to 30, 5 and 5 arbitrary units min⁻¹; vaporiser and ion transfer tube temperatures were both set to 300 °C; maximum injection time, 50 ms; AGC target: 100000; normalised AGC target: 25%; microscans, 10; RF-lens, 35%; data type, profile. Mass spectrometer calibration was performed using the Pierce FlexMix calibration solution (Thermo Scientific). MS data acquisition was carried out using the Xcalibur v. 4.3 software (Thermo Scientific). For data obtained at low resolution (3 or 7.5 K), only the major isotopic peak was detected, while analysis at high resolution (60 K) afforded the full isotopic spectrum (Supplementary Fig. 2).

### Tryptophan fluorescence quenching
All tryptophan fluorescence spectroscopy experiments were performed on a SAFAS Fluorescence Xenius Spectrophotometer (SAFAS, France) in a 2 mL quartz cuvette. The excitation wavelength was fixed at 295 nm and emission spectra were collected between 300–400 nm with a slit width of 2 nm. The temperature was maintained at 25 °C by an external thermostatic water circulator. To measure protein-ligand interactions, recombinant VirC, VirD, VirD E128A mutant and VirE at 5 μM were allowed to equilibrate in TE buffer (20 mM Tris-HCl pH 8.5, 2 mM EDTA) for 10 min under constant stirring, before being titrated with ligand solutions. The proteins were analysed against increasing concentrations of ligand (0–150 μM), depending on the specific ligand used. Data from two independent experiments were analysed using nonlinear regression, with application of the one site-specific binding model ($F = F_{max}*X/(K_d + X)$, where $X$ is the ligand concentration, $F$ is the fluorescence intensity, $F_{max}$ is the maximum specific binding and $K_d$ is the equilibrium binding constant) using SciDAVis v2.3.0.

### Circular dichroism measurements
Circular dichroism measurements were performed on a Chirascan CD (Applied Photophysics) in 100 mM NaPi, 150 mM NaF pH 8.0. Data were collected at 0.5 nm intervals in the wavelength range of 180–260 nm at 20 °C, using a temperature-controlled chamber. 30 μL of 100 μM ACP₅ₐ, ACP₅ₐ E6761A/L6764N and VirD were analysed in a 0.01 cm cuvette, while 100 μL of 100 μM VirD E128Q were analysed in a 0.1 cm cuvette. Each spectrum (Supplementary Fig. 1) represents the average of three scans, and sample spectra were corrected for buffer background by subtracting the average spectrum of buffer alone.

### Small-angle X-ray scattering (SAXS) data collection
SAXS data were acquired on the SWING beamline at the Synchrotron SOLEIL (France). The frames were recorded using an Eiger 4 M detector at an energy of 12 keV. The distance between the sample and the detector was set to 2000 mm for VirD, VirE, holo-ACP₅ᵦ–VirC, holo-ACP₅ᵦ–VirD, and holo-ACP₅ᵦ–VirE complexes, leading to scattering vectors q ranging from 0.0005–0.5 Å⁻¹. The scattering vector is defined as $4\pi/\lambda \sin\theta$, where $2\theta$ is the scattering angle. The protein samples were injected using the online automatic sample changer into a pre-equilibrated HPLC-coupled size-exclusion chromatography column (Bio-SEC 100 Å, Agilent), at a temperature of 15 °C.

The eluted fractions were delivered using an online purification system developed on the SWING beamline[47]. After equilibrating the column in the protein buffer (20 mM Tris-HCl pH 8.5, 300 mM NaCl, 5% glycerol), the buffer background was recorded (100 successive frames of 0.75 s). A 50 μL aliquot of the protein sample (at 5 mg mL⁻¹) was then injected, and complete data sets were collected. The protein

concentration downstream of the elution column was followed via the absorbance at 280 nm with an in-situ spectrophotometer. In lieu of analysing several protein concentrations within a standard range (e.g., 0.1–10 mg mL$^{-1}$), the coupling of data collection to a gel filtration column allowed analysis of multiple concentrations of protein within a single experiment, as many distinct positions within the elution peak were sampled during the course of the measurement (typically 50–100 frames are acquired)[47].

Following on from this, the dedicated in-house application FOXTROT was used to perform data reduction to absolute units, frame averaging, and solvent subtraction. Each acquisition frame of the experiment yielded a scattering spectrum, which was then analysed by FOXTROT to produce an $R_g$ (radius of gyration) as well as an $I(0)$ value (the $I(0)$ depends on the protein concentration at that position in the elution peak, as described by the Guinier law (approximation $I(q) = I(0)$ $\exp(-q^2 R_g^2 / 3)$ for $qR_g < 1.3$)). Notably, observing a constant $R_g$ for a significant proportion of the concentrations present in the gel filtration peaks showed that the measurements were concentration-independent, and thus that they were effectively carried out under conditions of infinite dilution.

All the frames exhibiting identical $R_g$ as a function of $I(0)$ were corrected for buffer signal and averaged. This step ensured that the obtained data reflected only the signal arising from the protein structure and not from intermolecular interactions. Finally, the distance distribution function $P(r)$ and the maximum particle diameter $D_{max}$ were calculated by Fourier inversion of the scattering intensity $I(q)$ using GNOM[48]. The SAXS data are presented in Supplementary Table 3.

## Molecular weights and oligomeric structures in solution from SAXS data

It is possible in principle to determine molecular weights from SAXS data using the $I(0)$ and the measured protein concentration. However, this method was not appropriate in our case, as the delay between exiting the gel filtration column and the SAXS data acquisition may have altered the concentrations. We therefore determined the molecular weights of the constructs using Bayesian Interference in PRIMUS[49]. SAXS data were recorded on VirD, VirE, as well as VirC, VirD and VirE complexed with holo-ACP$_{5b}$. SAXS data obtained on wild type VirC complexed with holo-ACP$_{5b}$ were directly compared with that calculated[21] from the crystal structure of the acetyl-ACP$_D$–CurD complex (PDB: 5KP6)[18]. OLIGOMER[50] was used to interpret the SAXS data obtained on holo-ACP$_{5b}$ in the presence of VirD in solution (for additional information, see Supplementary Fig. 6). A model of a trimer of VirE was generated using ColabFold[23] and CORAL[24], and rigid-body modelling of the holo-ACP$_{5b}$–VirE complex carried out using SASREF[25] (for additional information, see Supplementary Fig. 6). The quality of the models was determined using CRYSOL[21] to compare the fit between the theoretical scattering curves from atomic coordinates with experimental scattering curves, and judged using the discrepancy $\chi^2$, defined according to Konarev and colleagues[50].

## Crystallisation and X-ray data collection

Se-VirD was purified and stored in buffer (20 mM Tris-HCl pH 8.5, 300 mM NaCl, 5% glycerol) at a final concentration of 5 mg mL$^{-1}$. Holo-ACP$_{5b}$ was stored in buffer (20 mM Tris-HCl pH 8.5, 250 mM NaCl, 50 mM TCEP) at a final concentration of 20 mg mL$^{-1}$. Prior to crystallisation trials, sample homogeneity was checked by dynamic light scattering (DLS) using a Zetasizer NanoS (Malverne). Initial crystallisation hits were obtained using the Rigaku kit (Molecular Dimensions). The conditions consisted of 20% PEG 400, 20% PEG 800, 100 mM Tris-HCl, pH 7.5 for Se-VirD, while holo-ACP$_{5b}$–Se-VirD crystallised in 100 mM chloride calcium, 30% PEG 1500, 10% 2-propanol, 100 mM imidazole-HCl, pH 6.5.

Crystals grew in 10–15 days using the hanging drop method in Linbro® plates, with drops formed by mixing 2 μL of protein solution

(ratio 1:4 for the holo-ACP$_{5b}$–Se-VirD complex, 5 mg mL$^{-1}$ Se-VirD) with 1 μL of crystallisation buffer. Crystals were then soaked in crystallisation buffer containing 30% ethylene glycol prior to freezing in liquid nitrogen. X-ray diffraction data on Se-VirD and the holo-ACP$_{5b}$–Se-VirD complex were collected at the SOLEIL synchrotron on the Proxima2 beamline. The crystals belong to the P4$_1$2$_1$2 and H3 space groups, respectively (Supplementary Table 3). A complete MAD data set at four wavelengths was collected in order to solve the crystal structure of VirD. Data sets were indexed and integrated using XDS[51] and scaled by using pointless and aimless (CCP4 package).

## Structure determination and refinement

Initial phases were generated via SAD using the peak wavelength ($\lambda = 0.979260$ Å). Three high confidence Se sites were identified and refined by using NCS with Phenix AutoSol[52,53]. The figure of merit (FOM) from Phenix AutoSol is 0.32. Density modification and NCS were then used to improve the quality of the phases (FOM: 0.68 with a bias ratio of 1.36). The good quality of the electron density map allowed for building approximatively 80% of the backbone at 2.02 Å using Phenix AutoBuild[54]. The final model of WT VirD was built using ARP/wARP[55], followed by iterative cycles of manual rebuilding and refinement at 1.7 Å using COOT[56] and REFMAC5[57]. The structure of the holo-ACP$_{5b}$–VirD complex was solved by molecular replacement using a monomer of VirD as search model with the programme MOLREP in CCP4[58,59]. The contrasted solution with final CC of 0.7252 and Tf/sig of 27.17, consists of 2 monomers of VirD in the asymmetric unit. The initial model was then refined by rigid-body refinement at 3 Å followed by a restraint refinement at 2.1 Å resolution using REFMAC5 CCP4[57]. The excellent quality of the electron density maps allowed us to locate two extra electron densities in the $F_oF_c$ map corresponding to two ACP$_{5b}$ molecules in the asymmetric unit. The ACPs were then constructed manually in the electron density maps. Structure geometry was validated using the programme MolProbity[60]. The structures of VirD and holo-ACP$_{5b}$–VirD contain 99.26% and 97.91% of the residues in the allowed region of the Ramachandran plot, respectively, and no outliers (Supplementary Table 3). Figures were prepared using the programme PyMOL[61].

## Protein NMR data acquisition

All ACP protein samples were buffer exchanged via gel filtration into phosphate buffer (100 mM sodium phosphate pH 6.0, 1 mM EDTA, 1 mM TCEP), concentrated to 1 mM, and then 350 μL of the samples (including 10% D$_2$O) were loaded into 4 mm NMR tubes. All NMR data were recorded at 25 °C on a Bruker DRX600 spectrometer equipped with a cryogenic probe (Unité Mixte de Service (UMS) 2008 Ingénierie-Biologie-Santé en Lorraine (IBSLor)). Backbone and sequential resonance assignments were obtained by the combined use of 2D $^{15}$N–$^1$H and $^{13}$C–$^1$H HSQC spectra and 3D HNCA, HNCACB, CBCA(CO)NH, HNHA, HBHA(CO)NH, HN(CA)CO, and HNCO experiments. Assignments of aliphatic side chain resonances were based on 2D aromatic $^{13}$C–$^1$H HSQC, (HB)CB(CGCDCE)HE, (HB)CB(CGCD)HD and 3D (H)CC(CO)NH, H(CC)(CO)NH, CCH–TOCSY, and HCCH-TOCSY experiments (reviewed in ref. [62]). To collect NOE-based distance restraints for the structure calculations, 3D $^{15}$N NOESY-HSQC and $^{13}$C NOESY-HSQC were recorded on uniformly $^{13}$C,$^{15}$N enriched samples using a mixing time of 120 ms. NMR data were processed using Topspin 3.2 (Bruker) and were analysed using NMRFAM-SPARKY[63].

## Protein NMR structure calculations

CYANA 3.98 software[64] was used to generate initial structures, starting from manually-assigned NOEs. For this, the standard CYANA protocol was used, which consists of seven iterative cycles of calculations with NOE assignment carried out by the embedded CANDID routine, combined with torsion angle dynamics structure calculation[65]. During each cycle, 100 structures starting from random torsion angle values

were calculated with 15,000 steps of torsion angle dynamics-driven simulated annealing. A total of 1822, 1208 and 1763 NOE-based distances, and 110, 92 and 94 backbone angle restraints were used for structure calculation of the holo-ACP$_{5a}$, holo-ACP$_6$ and holo-ACP$_7$ domains, respectively (Supplementary Table 4). The angle restraints were obtained from $^{13}C\alpha$, $^{13}C\beta$, $^{13}C'$, $^{15}N$, $^1HN$, and $^1H\alpha$ chemical shifts using TALOS-N[66] with an assigned minimum range of ±20°. 4′-Phosphopantetheine-serine was created as a serine modified residue within the CYANA library using 4′-phosphopantetheine coordinates from the solution structure of holo-ACP PfACP from *Plasmodium falciparum* (PDB ID: 2FQ0)[67].

The second stage consisted of the refinement of the 50 lowest CYANA target function conformers by restrained molecular dynamic (rMD) simulations in Amber 14[68,69]. Phosphopantetheinyl serine library and force field parameters[70] were used for AMBER minimisation. The final representative ensembles correspond to the 20 conformers from each calculation with the lowest restraint energy terms. The structures of holo-ACP$_{5a}$, holo-ACP$_6$ and holo-ACP$_7$ contain respectively 98.6%, 94.4% and 97.1% in the most favoured region and 1.4%, 5.6% and 2.9% of the residues (non-glycine and non-proline) in the additionally allowed region of the Ramachandran plot. PRO-CHECK statistics were calculated using PROCHECK-NMR[71]. The proportion of residues in the most favoured/additionally allowed/generously allowed/disallowed regions of the Ramachandran plot for the ACPs are as follows: holo-ACP$_{5a}$ (97.1/2.9/0/0); holo-ACP$_6$ (94.3/5.7/0/0); holo-ACP$_7$ (92.4/7.1/0.1/0.4).

### Generation of *S. pristinaespiralis* pathway inactivation mutant

For construction of the pathway mutant, the pCRISPomyces-2 plasmid[31] was used for CRISPR-Cas9-based genome editing. Spacer sequences (Supplementary Data 1) were chosen using the online CRISPy-web software[72], and were generated by annealing two 24 nt oligonucleotides. Next, 1 kb homologous arms HAL and HAR were amplified by PCR, the pCRISPomyces-2 plasmid was linearised with the restriction enzyme *Xba*I (Thermo Fisher Scientific), and then assembly of the editing templates and the pCRISPomyces-2 plasmid was performed using the In-Fusion HD Cloning kit (Ozyme, France). Correct plasmid assembly was confirmed by diagnostic digestion and sequencing (Supplementary Fig. 9). Recombinant plasmids were introduced into *E. coli* 12567 (pUZ8002) by electroporation. Conjugation of plasmids into *Streptomyces* spores was performed using the protocol described previously[73]. Briefly, a single clone was used to inoculate a 5 mL pre-culture of LB medium supplemented with 25 mg L$^{-1}$ apramycin, 50 mg L$^{-1}$ kanamycin and 25 mg L$^{-1}$ chloramphenicol, and incubated at 37 °C, 180 rpm on a rotary shaker for 24 h. A volume of 1 mL of the pre-culture was then used to inoculate 25 mL of the same medium, and growth carried out to an $A_{600}$ of 0.5. The culture was then centrifuged at 1780 × $g$ for 10 min at room temperature, and the pellet resuspended in 25 mL of LB medium. This step was repeated twice, and the pellet was resuspended in 2 mL of ISP2 medium. In parallel, several replicates of 500 μL of ISP2 medium inoculated with a suspension of 10$^6$ spores of *S. pristinaespiralis* were heated at 50 °C for 10 min, and centrifuged at 4000 × $g$ for 1 min. Next, 500 μL of *E. coli* was added to each replicate suspension. The resulting suspensions were directly plated onto RP agar plates containing 10 mM MgCl$_2$, and then incubated at 30 °C for 7 days. After conjugation, clearance of the plasmid was accomplished by repeated high-temperature cultivation (37 °C) for 2–3 days, followed by replica plating on selective and nonselective plates to confirm restoration of apramycin sensitivity. Apramycin-sensitive colonies were then picked into liquid ISP2 medium (4 g L$^{-1}$ yeast extract, 4 g L$^{-1}$ dextrose, 10 g L$^{-1}$ malt extract adjusted to pH 7.3 with NaOH) for genomic DNA isolation using the Wizard Genomic DNA Purification Kit (Promega). Genomic modifications were confirmed by PCR and sequencing of the modified regions (Supplementary Fig. 9).

### Analysis by HPLC-MS of *S. pristinaespiralis* wild type, the *S. pristinaespiralis* pathway inactivation mutant and *S. virginiae*

Cultures were extracted twice with ethyl acetate (v/v). When present, XAD-16 resin was harvested by sieving, and also extracted twice with ethyl acetate (v/v). The solvent was removed by evaporation, the extracts resuspended in 1:1 ACN/water (v/v) and then the sample was passed through a 0.4 μm syringe filter. HPLC-MS analysis was performed in positive and/or negative electrospray mode (ESI+/−) on the Thermo Scientific Orbitrap ID-X Tribrid Mass Spectrometer using an Alltima™ C18 column (2.1 × 150 mm, 5 μm particle size) at 25 °C (flow rate, 0.2 mL min$^{-1}$) or an Interchim Uptisphere C18 column (2.1 × 150 mm, 5 μm particle size) (Supplementary Fig. 10 only). Separation was carried out with Milli-Q water containing 0.1% formic acid (A) and acetonitrile containing 0.1% formic acid (B), using the following elution profile: 0–48 min, linear gradient 5 – 95% solvent B; 48–54 min, constant 95% solvent B; 54–60 min, constant 5% solvent B. In the case of the comparative analysis of *S. virginiae* and *S. pristinaespiralis* (Supplementary Fig. 12), and to obtain clean MS$^2$ data on metabolite **3** (Fig. 5), separation was carried out on a Phenomenex Luna Omega Polar C18 column (3 × 100 mm, 5 μm particle size) with solvent A and B as above, using the following elution profile: 0–20 min, linear gradient 1–99% solvent B; 20–25 min, constant 99% solvent B; 25–25.1 min, linear gradient 99–1% solvent B; 25.1–31 min, constant 1% solvent B. Mass spectrometry operating parameters were as described above. Metabolite yields (Supplementary Table 5) were estimated by generating a calibration curve using commercially-available virginiamycin M **1** (Sigma-Aldrich), over the concentration range of 0.00128–20 mg L$^{-1}$ (10 μL of each sample was injected). This approach afforded a linear correlation between the quantity of metabolite and the respective integrated peak area in the extracted ion chromatogram (EIC) (the areas of the peaks corresponding to the parental ions [M + H]$^+$ were used systematically) (Supplementary Fig. 11). For analysis of metabolite yields in extracts (Supplementary Table 5), following conversion of peak areas to titres, the results were divided by 200 to correct for the enrichment of the sample during preparation, as the extracts from 20 mL of culture were resuspended in 100 μL of solvent prior to HPLC-MS analysis (as with the standard, 10 μL of each sample was injected).

### Reporting summary

Further information on research design is available in the Nature Portfolio Reporting Summary linked to this article.

## Data availability

Crystal structures of VirD and the holo-ACP$_{5b}$–VirD complex have been deposited in the Protein Data Bank with their respective diffraction data under accession codes 8AHZ and 8AHQ, respectively. Coordinates and chemical shifts for the NMR structures of holo-ACP$_{5a}$, holo-ACP$_6$ and holo-ACP$_7$ have been deposited in the Protein Data Bank with accession codes 8A7Z, 8AIG, and 8ALL, and in the Biological Magnetic Resonance Bank with accession codes 34739, 34743 and 34744, respectively. Raw SAXS and HPLC-MS data have been deposited in the data repository DOREL (DOnnées de la REcherche Lorraines) [https://dorel.univ-lorraine.fr/] with accession code https://doi.org/10.12763/GYAWHI. The remaining data supporting this study are included in the Supplementary Information. All biological materials are available from the authors upon request. Source data are provided with this paper.

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

## Acknowledgements

We acknowledge financial support from the Agence Nationale de la Recherche (grant numbers ANR-11-JSV8-003-01, PKS-PPIs; ANR-16-CE92-0006-01, PKS STRUCTURE; and, ANR-20-CE93-0002-01, PKSOx to K.J.W.), the Université de Lorraine and the Centre National de la Recherche Scientifique (CNRS). We also acknowledge J. Davison for help with the molecular biology, Omar A. Rifi for assistance with protein production and modification, and W. Shepard and M. Savko (Soleil Synchrotron, Proxima2) as well as J. Perez and A. Thureau (Soleil Synchrotron, Swing) for help with data acquisition. Crystal screening for diffraction quality and acquisition of NMR data were carried out on the Plateforme de Biophysique et Biologie Structurale (B2S) (IBSLor, UMS2008, CNRS-UL-INSERM). Analytical chemistry was performed on the Structural and Metabolomics Analyses Platform (PASM), SF4242, Université de Lorraine, EFABA, Vandœuvre-lès-Nancy, France.

## Author contributions

A.G., K.J.W. and S.C. designed the study and carried out comparative sequence analysis. S.C. and B.C. designed and performed the molecular biology experiments. S.C. expressed and purified recombinant proteins, and S.C. and B.C. generated modified versions. S.C. performed the biophysical analyses, carried out the X-ray crystallography and SAXS analysis with A.G., and engineered the pathway inactivation with help from C.J. B.C. solved and analysed the ACP NMR structures. C.P. carried out the HPLC-MS experiments, and along with R.J.C., helped K.J.W. with data analysis and interpretation. All authors discussed the results. K.J.W., A.G. and S.C. wrote the manuscript, with input from BC.

## Competing interests

The authors declare no competing interests.
