## [Peer Review File · Nature Communications]

Decrypting the programming of β -methylation in virginiamycin M biosynthesisREVIEWER COMMENTS

Reviewer #1 (Remarks to the Author):

The report by Weissman and co-workers provides a detailed study of the structural features of ACP domains that govern recognition by HMGS β -branching cassette enzymes. This work extends earlier studies that at the time predicted the existence of tryptophan flag which was consistent with (most) sequences available at the time. Although outliers were identified, including virginiamycin, nearly a decade later many more sequences are available that do not conform to this model. In summary, this key area is ripe for revisiting.

The present study now takes the ambiguities identified in several didomain pairs and presents key biophysical and structural data that now strongly supports a model in which one ACP of a didomain pair is involved in chain extension only whilst only the second is able to β -branch. This 'in series' behaviour goes against current dogma that tandem ACPs act solely in parallel and ascribes Vir 5a and 5b separate functions. Moreover, the demonstration that β -branching may escape spatio-temporal kinetic control at a secondary site (ACP7) is particularly interesting as is demonstration that β -branching might be controlled in these and related systems.

This alone would be worthy of a high impact paper, but the study also includes a plethora of NMR structure determination, SAXS, X-ray studies, SDM, gene-editing and biophysical data that dissects the precise mode of molecular recognition between acceptor ACP and branching cassette enzymes. In this regard, a further notable highlight of this work is the trapping of a high-resolution ACP-ECH1 complex. ACP/PKS complexes are still rare and therefore any new information here is valuable. In particular this is the first example of an ACP-ECH complex, again of significance given lack of information of these protein complexes.

The data are robust but the authors should consider the following;

1) The fitting to the SAXS data for the VirE-ACP5b complex. The χ^2 of >5 indicates a very poor fit to this data. Whilst fitting a VirE-Acp5b alphafold model generates an improvement over fitting to VirE alone, the authors may want to consider adding a caveat to the main text to highlight the trend but limitations of the dataset currently. Potentially further fitting using EOM to account for flexible regions would further enhance this fit.

2) The raw SAXS data for VirD and VirE should also be shown in extended Figure 3. Also there is some confusion in the SI table 3 – there are not stats for holo-ACP5b-VirE OR this is a typo in the fourth column labelled holo-ACP5b-VirD? With an X-ray structure in hand for holo-ACP5b-VirD, this would also have been very useful to have looked at by SAXS for comparison and fitting? Did the authors acquire this data?

3) Third, the key finding that the subtle electrostatic distribution controls β -branching could be presented with a wider sequence analysis. At present this is restricted to two very closely related systems where orthogonality within the PKS may be maintained in this way. What is less clear, and extended data figure 4b is a bit unhelpful, is how these trends extend to other systems? Is this charge orthogonality generalisable and what can be predicted elsewhere?

In summary this is an elegant study that opens up the structural stage for engineering β -branching components of Trans-AT pathways. This will have a broad interest across natural product and protein engineering and biochemistry fields.

Minor Suggestions

1) Figure 1- in B, include key to enzymes above arrows to make this more accessible, eg KS0 above VirB etc. Also figure is very small and barely legible even at printed A4 size.

- 2) Also Figure 1 – part c) no rationale for why Sna is shown here and not Vir? Also is this a bit premature to have in the introductory figure.
- 3) Check figures – multi-panel figures do have objects labelled in the legend but adding labels above structures helps the reader a lot – eg Figure 4 add ACP labels above each structure. Suggest same for extended data Figure 2.
- 4) Introduction lines 46-51. A comment that several outliers including leinamycin and virginiamycin had been highlighted as ambiguous and lacked the Trp flag, would clarify why the recognition mechanism remains obscure.
- 5) No comment is made on the current mutational analysis and reconciling previous studies where mutation of residues in helix 3 in an acceptor ACP impacted metabolite production (Haines et al.). Could more than one binding mode be operational (to the HCS for example) or was helix 3 important for other recognition features?
- 6) Figure 5 needs attention – text too small and needs tidying.
- 7) Extended Figure 7 – black text on purple hard to see and can available space be used to make structures larger as these are complex and text difficult to discern. Red and orange very difficult to tell apart. One branch labelled red ions but corresponding orange ions label. Also New compound is compound number 3.

Minor errors including but not limited to;

- 1) Line 48 sp. gouverns
- 2) Line 60 sp. Phophopantetheine
- 3) Extended Data Figure 4b – rhombus is used to highlight W, not an arrow. Also line 142 first refers to T6849 but this is T6850 (line 126 for example) Which sequence numbering is being used in 4a? This should be stated. Also Line 911, ‘as these formed the basis...’ implies only these two sequences were considered as a basis for this model. Infact these and known ACPs from numerous pathways were considered and W was presented a strong predictor but clearly there were also exceptions. Also, the hearts/club etc symbols are very difficult to see or obscured (eg overlap with number 20 on top row). Is there a need for the yellow shading and boxes? Could the pile-up be carefully edited to better highlight key differences between interface residues and charged groups? I found this presentation very unhelpful. A separate pileup of 5a and 5b together would also be a simple way to illustrate some of the key differences.
- 4) Line 278 – cites reference 29 as supporting solely in parallel behaviour – the title actually highlights in series behaviour of a tri-domain ACP construct which appears to contradict the authors statement.

Reviewer #2 (Remarks to the Author):

In this manuscript, Collin et al. investigate the structural elements that enable ACP recognition and beta-methylation in module 5 of the virginiamycin PKS-NRPS. The two ACPs in this module lack the Trp residue previously shown to indicate ACPs that are recognized by beta-methylation enzymes in other systems. All of the enzymes involved in beta-methylation (VirC-E), not just the first (VirC), preferentially bind to the relevant ACP, ACP5b. The authors use structural biology to provide evidence that the electrostatic character of ACP5b enables recognition. Thus, the authors data argue against the previous hypothesis that the orientation of helix alpha3 in ACPs targets them for the beta-methylation enzymes. The authors show that another ACP (ACP7) in the virginiamycin PKS-NRPS preferentially bound by VirC-E. They identify a doubly methylated virginiamycin analog in a parallel *Streptomyces* species and provide evidence for its proposed structure.

It is unclear how general the findings of this paper are; that is, it is unclear whether the structural elements that the authors identify are relevant for other ACPs that are targeted for beta-methylation or whether the authors are investigating a specialized case. It is unclear whether the authors would have been able to identify ACP7 as a target for beta-methylation based on structure alone (that is, without binding data). The structural characterization of the doubly methylated virginiamycin is also questionable.

Fig. 2c is not well depicted and is difficult to digest.

Does *S. virginiae* make 3?

Are the intermediates detected in ED Fig. 5b also found in the deletion mutant? Are the intermediates detected in ED Fig. 5b labeled by the deuterium-labeled amino acids as one would expect?

For ED Fig. 7, part c, identification of smaller fragments of 3 would help to verify its structure. In part d, the 260.0917 fragment does not seem reasonable (somehow the oxazole and the carboxylate are fused together with removal of proline?). In part d, it seems that ring opened forms of all of these product ions are more likely than ring closed forms.

For ED Fig. 8, metabolites 1 and 2, part b, why is +1D not seen in the spectrum?

For ED Fig. 8, metabolite 3, part c, why is +4D not seen in the spectrum?

The authors make this statement: "...while the presence of substrate analogues did not increase but moderately diminished affinity. The latter result, which implies substrate tolerance, ..." The logic behind this statement is unclear.

Reviewer #3 (Remarks to the Author):

The authors provide insights into the mechanism underlying β -methylation in trans-AT PKS systems, thereby using virginiamycin biosynthesis as an example. By structural biology, they reveal the basis for substrate choice by recognizing the respective domains.

In general, the analyses performed are sound and the conclusions seem to be reasonable for the system used. However, it might be of interest to test if this is a general theme applicable to other ACP β -methylation enzyme "pairs". If I got the authors right, the structure of the ACPs is quite conserved. Crystal structures exist, and if not alpha fold should be able to provide good predictions. For β -methylation enzymes, also structures were generated. Hence, would it be possible to model this on a bigger level? This analysis should be done to deduce (at least in silico) if the residues that were shown to be essential in complex formation are crucial in general. (Being aware that the authors wrote: "understanding the detailed role played by these residues in the interaction awaits higher resolution structural information".)

Some points that might help to make the manuscript more clear to the reader:

Line 126: The residues mentioned here are not specifically depicted in Figure 2c. Would it be possible to do so (thereby, of course keeping the readability of the figure)?

Line 190: Even though in the context it becomes clear, the sentence is not directly clear which charge is disfavoring complex formation. In Figure 4 the label of the residues is hard to read (especially black on dark blue). It should be referred to a figure to clarify this point.

Line 234: Please elaborate on why "this modification is reduced under native biosynthetic conditions".

Line 281: The speculation that holo-ACP5b is not an efficient substrate for malonylation by the trans-acting AT could be easily tested using the methodology described in the manuscript. This should be experimentally be done. In addition, the complex formation between ACP, β -methylation cassette and trans-AT can be tested (maybe using binding affinities as done by the authors).

Line 305: The authors propose that deblocking latent chemistries would diversify polyketide structures. To me, it remains a bit unclear how that should be done. In the study, only minor side products had been identified. However, this was not triggered in a rational way.

Line 919: In the Extended Data Fig. 5 (and the others) the reads of the chromatograms are always given on a scale from 0 to 100. It would be interesting to see the absolute values to judge about the abundance of a signal. (Maybe this should be indicated by the area, which is not available for all. Maybe comment on this)

Reviewer #4 (Remarks to the Author):

Collin and coworkers report a fundamental study on the selectivity of beta-alkylation in virginiamycin M biosynthesis via the detailed characterization of protein structures and protein-protein interactions of VirC/D/E and ACPs. The authors demonstrate that ACP5b is the scaffold for beta-alkylation, and surprisingly, ACP7 also provides another possible scaffold. The authors further show the existence of a trace amount of the doubly beta-methylated product in the cells. In addition, the authors succeed in proposing some key mechanisms for the distinguishment of acyl-ACP substrates to be beta-alkylated. The manuscript is well written, the data are very solid, and the story is fascinating. This study is important for understanding the basal mechanisms of beta-alkylation of polyketides, and it will open the new way for modifying the chemical structures of polyketides by controlling beta-alkylation. Unfortunately, however, some description and figures are not clear to understand. Extended Data Figures are messy throughout. This manuscript can be accepted after some revisions as below.

Major comments.

- 1) Introduction. It may be enough for the PKS scientists, but it is hard for general readers. I suggest the authors to include basic information, e.g., polyketides are... and the fact that beta-alkylation resembles the mevalonate pathway.
- 2) Introduction, Lines 52-67. This paragraph is redundant.
- 3) In the manuscript, the authors use the term "VirC-E" as the meaning of "VirC, VirD, and/or VirE". However, this confuses the readers because it also seems like "VirC-VirE complex". Particularly, in Line 79, "binding of VirC-VirE to the holo-ACP5a-ACP5b" is so complicated.
- 4) Line 267. I felt a bit uncomfortable with "even when excised from its modular context". Actually, ACPs were truncated from the original megaenzymes in this study. Then, do the authors think that the truncated ACPs have different conformations compared to those in the original megaenzymes?
- 5) Line 269. "The gate-keeping function within the cassette is therefore not limited to the HMGS VirC". Difficult to understand the connection with the previous sentence.

Minor comments.

- 1) Fig. 1a. Modules 3, 8, and 10 are shown with gray bars. Do these show that these modules are NRPSs? Module 9 does not harbor its growing chain, so does this show that this module is not involved in chain elongation? These explanations should be included in the figure caption.
- 2) Fig. 1b. Left structure is designated as VirAB. Is this correct?
- 3) Fig. 4. Letters of amino acid residue numbers are not clear.
- 4) Extended Data Table 1. This table includes much important information. Nevertheless, it is a little bit bothersome to compare the results. My suggestion would be: ACPs are shown in the first column, then modifications (apo-, holo-, etc.) are shown in the subcolumn. The authors do not necessarily have to follow this, but can the authors reorganize the Table?
- 5) Extended Data Fig. 1a. Stacking gel can be cut. The index of MW markers is out of position.
- 6) Extended Data Fig. 4. Suits are too small and hard to see. NCBI accession numbers for all proteins are required.
- 7) Supplementary Figure 4. The R-2 value is 1.00, but generally it means that all points are on the line. In this case, it is due to the number of digits. Can the authors increase the digits to show that the value is not actually 1.00? Then, the caption should also be changed.

RESPONSE TO REVIEWERS

Reviewer #1 (Remarks to the Author):

The report by Weissman and co-workers provides a detailed study of the structural features of ACP domains that govern recognition by HMGS β -branching cassette enzymes. This work extends earlier studies that at the time predicted the existence of tryptophan flag which was consistent with (most) sequences available at the time. Although outliers were identified, including virginiamycin, nearly a decade later many more sequences are available that do not conform to this model. In summary, this key area is ripe for revisiting.

The present study now takes the ambiguities identified in several didomain pairs and presents key biophysical and structural data that now strongly supports a model in which one ACP of a didomain pair is involved in chain extension only whilst only the second is able to β -branch. This 'in series' behaviour goes against current dogma that tandem ACPs act solely in parallel and ascribes Vir 5a and 5b separate functions. Moreover, the demonstration that β -branching may escape spatio-temporal kinetic control at a secondary site (ACP7) is particularly interesting as is demonstration that β -branching might be controlled in these and related systems.

This alone would be worthy of a high impact paper, but the study also includes a plethora of NMR structure determination, SAXS, X-ray studies, SDM, gene-editing and biophysical data that dissects the precise mode of molecular recognition between acceptor ACP and branching cassette enzymes. In this regard, a further notable highlight of this work is the trapping of a high-resolution ACP-ECH1 complex. ACP/PKS complexes are still rare and therefore any new information here is valuable. In particular this is the first example of an ACP-ECH complex, again of significance given lack of information of these protein complexes.

The data are robust but the authors **should consider** the following;

1) The fitting to the SAXS data for the VirE-ACP5b complex. The χ^2 of >5 indicates a very poor fit to this data. Whilst fitting a VirE-Acp5b alphafold model generates an improvement over fitting to VirE alone, the authors may want to consider adding a caveat to the main text to highlight the trend but limitations of the dataset currently. Potentially further fitting using EOM to account for flexible regions would further enhance this fit.

Response:

We thank the reviewer for pointing out the limitations of our initial analysis. We have addressed this criticism by a multi-step procedure, which is now described in **Extended Data Fig. 3**, panel **d**. Briefly, a substantially improved fit ($\chi^2 = 1.89$) was obtained by truncating VirE to remove the helix α_{10} regions which likely undergo induced folding during complex formation, as well unstructured regions of ACP_{5b}. We did attempt to implement EOM to further improve the fit, but were unsuccessful. Indeed, as now noted in the legend to **Extended Data Fig. 3**, multiple factors complicate an EOM-based analysis of this complex, including the fact that the VirE region which becomes ordered participates in the very interfaces that EOM is attempting to model by rigid-body docking, and the dominance of the SAXS signal by the larger VirE trimer. In any case, the overall conclusion remains unchanged – that the *holo*-ACP_{5b}-VirE complex resembles that of *holo*-ACP_{5b}-VirD, with the ACPs positioned at the interfaces between VirE monomers.

2) The raw SAXS data for VirD and VirE should also be shown in extended Figure 3. Also, there is some confusion in the SI table 3 – there are no stats for *holo*-ACP_{5b}-VirE OR this is a typo in the fourth column labelled *holo*-ACP_{5b}-VirD? With an X-ray structure in hand for *holo*-ACP_{5b}-VirD, this would also have been very useful to have looked at by SAXS for comparison and fitting? Did the authors acquire these data?

Response:

We are unclear what the referee intended to say here concerning the raw data, as the SAXS curves are systematically presented. Concerning **Supplementary Information Table 3**, the referee is correct in that an error

was inadvertently introduced. The column does indeed report the data obtained on the *holo*-ACP_{5b}-VirE complex, and this error has now been corrected. We did in fact acquire SAXS data on the *holo*-ACP_{5b}-VirD complex which were not included in the previous version, but which have been added (**Extended Data Fig. 3c**). This analysis showed that the sample contained a 7:3 mixture of VirD and the *holo*-ACP_{5b}-VirD complex. Our attempts to boost the proportion of complex by increasing the relative ACP concentration were unsuccessful. However, considering this mixed population using OLIGOMER substantially improved the fit to the SAXS data (from $\chi^2 = 10.3$ based on the crystal structure of *holo*-ACP_{5b}-VirD complex (PDB ID: 8AHQ), to $\chi^2 = 1.7$ based on the 7:3 mixture).

3) Third, the key finding that the subtle electrostatic distribution controls β -branching could be presented with a wider sequence analysis. At present this is restricted to two very closely related systems where orthogonality within the PKS may be maintained in this way. What is less clear, and extended data figure 4b is a bit unhelpful, is how these trends extend to other systems? Is this charge orthogonality generalisable and what can be predicted elsewhere?

Response:

We thank this and reviewers 2 and 3 for these constructive comments. To address this question, we analysed ACP/VirD homologue pairs from several sets of evolutionarily related systems, reasoning that within families, the key specificity elements were more likely to be conserved. The target systems included the bacillaene (Bae)/Pks *trans*-AT PKSs and curacin (Cur)/jamaicamide (Jam) *cis*-AT PKSs, for which biochemical data were available concerning the mode-of-operation of the β -methylation ACPs (all were described to act in-parallel (10.1073/pnas.0603148103, doi: 10.1038/nature07870)). We also focused investigated the pederin family of PKSs, the largest set of related *trans*-AT systems described to date.

This analysis revealed several important observations, which indeed support the generality of our findings:

1. Strict conservation (physical character if not residue identity) among the key specificity determinants of the Bae/Pks and Cur/Jam ACPs present in modules targeted for β -methylation, which is fully in accord with their reported in-parallel function.
2. With only one exception, the finding that the two α 1- α 2 loop residues which we propose contribute to Ppant positioning/domain electrostatic surface features never occur together in any ACPs located in non- β -methylation modules, and that even the presence of one of the two residues is atypical. This observation is consistent with the role of these residues as specificity determinants, even if the detailed mechanism by which specificity is conferred may differ among systems.
3. Strong conservation among the homologues of VirD from the above systems of electrostatic interactions with ACP_{5b} mediated by three Arg residues. The observed positively-charged amino acids in the homologues either occupy the equivalent position as in VirD, or locations which according to the *holo*-ACP_{5b}-VirD structure, would also allow them to interact productively with the target ACP structural elements.

These observations have been incorporated into the text, and the associated sequence alignments are presented in new **Supplementary Figs. 8–13**.

In summary this is an elegant study that opens up the structural stage for engineering β -branching components of *Trans*-AT pathways. This will have a broad interest across natural product and protein engineering and biochemistry fields.

Minor Suggestions

1) Figure 1- in B, include key to enzymes above arrows to make this more accessible, eg KSO above VirB etc. Also figure is very small and barely legible even at printed A4 size.

Response:

As requested, we have incorporated additional explanatory features into the figure, and increased the image and font sizes.

2) Also Figure 1 – part c) no rationale for why Sna is shown here and not Vir? Also is this a bit premature to have in the introductory figure.

Response:

In the interest of space and to permit a direct comparison of the Vir and Sna systems, we elected to include Sna in this initial figure. If this is not satisfactory, we will include the Sna assembly line as an additional Supplementary figure.

3) Check figures – multi-panel figures do have objects labelled in the legend but adding labels above structures helps the reader a lot – eg Figure 4 add ACP labels above each structure. Suggest same for extended data Figure 2.

Response:

As requested, we have added ACP labels to **Fig. 4**. Concerning **Extended Data Fig. 2**, we note that each of the spectra had been labelled according to the ACP being analysed in the original version. We have, however, altered the colour code, so that the spectra arising from each ACP are shown in the same colour (i.e. Vir ACP_{5a} = red).

4) Introduction lines 46-51. A comment that several outliers including leinamycin and virginiamycin had been highlighted as ambiguous and lacked the Trp flag, would clarify why the recognition mechanism remains obscure.

Response:

We thank the reviewer for pointing out this omission. We have modified the section to read as follows:

However, several ACP domains targeted for β -methylation were identified which lacked the conserved Trp (e.g. in the virginiamycin (Vir) M and leinamycin PKSs)⁷, calling into question the proposed recognition mechanism.

5) No comment is made on the current mutational analysis and reconciling previous studies where mutation of residues in helix 3 in an acceptor ACP impacted metabolite production (Haines et al.). Could more than one binding mode be operational (to the HCS for example) or was helix 3 important for other recognition features?

Response:

We thank the reviewer for this comment and apologize for the omission. On the other hand, as Haines *et al.* investigated the ACP_A/HMGS interaction in mupirocin biosynthesis, we felt it was more relevant to discuss this result in the context of the comparison between the SAXS data obtained on the *holo*-ACP_{5b}-VirC (HMGS) complex and that predicted based on the acetyl-ACP_D-CurD crystal structure (pages 3–4). The acetyl-ACP_D-CurD structure indeed identified ACP_D helix α 3 as an interface element, while Haines, *et al.* probed the involvement of this same helix in the interaction between an ACP_A and the CurD homologue MupH in the mupirocin pathway. The resulting drop in mupirocin production upon mutation of helix α 3 is consistent with the idea that helix α 3 is a motif used by HMGS enzymes generally to recognize both ACP_D and ACP_A partners. These observations have been added to page 3, as follows:

Nonetheless, comparison of small-angle X-ray scattering (SAXS) data obtained on wild type VirC complexed with *holo*-ACP_{5b}, with that calculated¹⁸ from the crystal structure of the acetyl-ACP_D-CurD complex (PDB: 5KP6)¹⁵, revealed a remarkable fit between the experimental and theoretical scattering curves ($\chi^2 = 1.52$) (**Extended Data Fig. 3, Supplementary Table 3**). This result shows that the overall structures are similar, implying that HMGS recognition of both ACP_D and ACP_A partners involves common structural elements. In the acetyl-ACP_D-CurD case¹⁵, the interface encompasses the entirety of helix α 2, the loop α 2- α 3 and helix α 3, as well as a key orientational interaction between the Ppant phosphate and CurD Arg33¹⁵. Further evidence that helix α 3 plays a role in HMGS recognition of ACP_A was provided by mutation of an ACP_A of the mupirocin PKS, which resulted in a 3–10 fold decrease in mupirocin production *in vivo*¹⁰.

We would argue that the helix α 3 mutagenesis results are less relevant to the mutagenesis carried out on Vir ACP_{5b} to probe its interaction with VirD. Nonetheless, we have now pointed out the fact that ACP_{5b} helices α 2 and α 3 are likely used to mediate contacts with both VirC and VirD (page 4):

Concerning the ACP, the interaction involves the C-terminal portion of helix $\alpha 1$, the adjacent loop ($\alpha 1-\alpha 2$) and the N-terminal regions of helices $\alpha 2$ and $\alpha 3$. Thus, while complex formation with VirC and VirD involves shared ACP elements (helices $\alpha 2$ and $\alpha 3$), the overall interaction surfaces are distinct.

We also note in response to the reviewer that we did not choose to mutate the helix $\alpha 3$ residues of ACP_{5b} to test their roles, as we were particularly focused on amino acids that could be used to distinguish between ACP_{5b} and ACP_{5a}, and the helix $\alpha 3$ residues which mediate recognition (Y6895 and D6896, ACP_{5b} numbering) are identical (Extended Data Fig. 4).

6) Figure 5 needs attention – text too small and needs tidying.

Response:

As requested, we have revised Fig. 5 by removing extraneous peak labels and by increasing the font size.

7) Extended Figure 7 – black text on purple hard to see and can available space be used to make structures larger as these are complex and text difficult to discern. Red and orange very difficult to tell apart. One branch labelled red ions but corresponding orange ions label. Also New compound is compound number 3.

Response:

In response to the referee's comments, we have extensively revised this figure: ions highlighted in the mass spectra are coloured instead of boxed, and the figure sizes have been increased. In response to referee 2 (comment 7), we have also presented new fragmentation schemes accounting for additional ions observed in both metabolites 1 and 3.

Minor errors including but not limited to:

1) Line 48 sp. Gouverns

Response: Corrected.

2) Line 60 sp. Phophopantetheine

Response: Corrected.

3) Extended Data Figure 4b – rhombus is used to highlight W, not an arrow (it is an arrow, but inverted and not clear).

Also line 142 first refers to T6849 but this is T6850 (line 126 for example)

Response: Corrected.

Which sequence numbering is being used in 4a? **This should be stated.**

Response:

The sequence numbering corresponds to that of VirA ACP_{5b}. This is now stated in the legend.

Also Line 911, 'as these formed the basis...' implies only these two sequences were considered as a basis for this model. In fact, these and known ACPs from numerous pathways were considered and W was presented a strong predictor but clearly there were also exceptions.

Response:

This sentence has been changed to the following: Both MupA ACP_{3a} and ACP_{3b} are included as representative sequences upon the W flag model was based⁷ (position highlighted in red). (Note, the nomenclature of the mupirocin ACPs has been changed to reflect that in the Haines, *et al.* paper)

Also, the hearts/club etc symbols are very difficult to see or obscured (eg overlap with number 20 on top row). Is there a need for the yellow shading and boxes? Could the pile-up be carefully edited to better highlight key differences between interface residues and charged groups? I found this presentation very unhelpful. A separate pileup of 5a and 5b together would also be a simple way to illustrate some of the key differences.

Response:

We have modified **Extended Data Fig. 4** to address the referee's concerns. Panel **a** now comprises an alignment between Vir ACP_{5a} and ACP_{5b}. Panel **b** includes all of the tested ACPs from the Vir system, while **c** illustrates newly-identified *trans*-AT PKSs whose β -methylation ACPs do not incorporate the characteristic W flag. Finally, panel **d** compares all of the ACPs from the Vir and Sna systems.

4) Line 278 – cites reference 29 as supporting solely in parallel behaviour – the title actually highlights in series behaviour of a tri-domain ACP construct which appears to contradict the authors statement.

Response:

We chose to include reference 29 as it did report in-parallel behaviour for ACPs involved in β -modification. The referee is correct in that the data also supported in-series action of ACP domains from the same system, but as these are involved in mediating standard PKS reactions *in cis*, their relevance to the present study is unclear. Nonetheless, if the referee prefers, we can remove this reference.

Reviewer #2 (Remarks to the Author):

In this manuscript, Collin *et al.* investigate the structural elements that enable ACP recognition and beta-methylation in module 5 of the virginiamycin PKS-NRPS. The two ACPs in this module lack the Trp residue previously shown to indicate ACPs that are recognized by beta-methylation enzymes in other systems. All of the enzymes involved in beta-methylation (VirC-E), not just the first (VirC), preferentially bind to the relevant ACP, ACP_{5b}. The authors use structural biology to provide evidence that the electrostatic character of ACP_{5b} enables recognition. Thus, the authors data argue against the previous hypothesis that the orientation of helix alpha3 in ACPs targets them for the beta-methylation enzymes. The authors show that another ACP (ACP₇) in the virginiamycin PKS-NRPS preferentially bound by VirC-E. They identify a doubly methylated virginiamycin analog in a parallel *Streptomyces* species and provide evidence for its proposed structure.

1) It is unclear how general the findings of this paper are; that is, it is unclear whether the structural elements that the authors identify are relevant for other ACPs that are targeted for beta-methylation or whether the authors are investigating a specialized case.

Response:

Please see response to Referee 1, point 3).

2) It is unclear whether the authors would have been able to identify ACP₇ as a target for β -methylation based on structure alone (that is, without binding data).

Response:

No, we could not have predicted ACP₇ targeting, and indeed were surprised by the result given that β -methylation had been assumed to be highly specific. In contrast, with binding data in hand, we were able to explain the observation based on the globally similar electrostatic surface features of ACP_{5b} and ACP₇, and the striking differences with those of ACP_{5a} and ACP₆. Nonetheless, we are not yet at the stage of being able to confidently predict whether a given ACP can be recognised. Answering this question will require studying a large number of ACPs from additional *trans*-AT PKSs, characterising both their partner binding and surface potentials – research which is beyond the scope of the present work.

3) The structural characterization of the doubly methylated virginiamycin is also questionable.

Response:

As further detailed in the arguments below, the MS² and amino acid incorporation data are fully consistent with the proposed structure of **3**. No further structural proof (i.e. by NMR) is possible, given the minute yields of **3**.

4) Fig. 2c is not well depicted and is difficult to digest.

Response:

We have improved this figure by increasing the contrast between the various domains, and by removing the second layer of contouring of the Ppant electronic density.

5) Does *S. virginiae* make **3**?

Response:

We thank the reviewer for this query which has prompted further investigation. Indeed, analysis for metabolites **1–3** in extracts of *S. virginiae* grown under the same conditions as *S. pristinaespiralis*, demonstrates that all three compounds are present. The yields of **1** and **2** from *S. virginiae* are 30% and 17%, respectively, those from *S. pristinaespiralis*. Concerning **3** from *S. virginiae*, it is produced at ca. 1000-fold lower yield than **1** in the same strain, demonstrating that the control of β -methylation is tighter. We have added these results to the text, as follows:

Analysis of *S. virginiae* also demonstrated the presence of **3** in addition to **1** and **2**, but at ca. 1000-fold lower yield than **1** from the same strain (**Supplementary Fig. 6**).

Given the prevailing view in the literature that β -modification occurs with high fidelity⁷, we were surprised to observe that ACP₇ is also efficiently recognised by the three cassette enzymes *in vitro*, an interaction which translates *in vivo* in two strains of *Streptomyces* into a previously-unidentified Vir M analogue **3** bearing a second β -methyl group. Notably, titres of **3** at ca. 0.1–1% of those of **1**, are on par with amounts of polyketides typically obtained by PKS genetic engineering⁴.

Nonetheless, control of β -methylation in *S. virginiae* is evidently tighter than in *S. pristinaespiralis*, as *S. virginiae* produces proportionally lower amounts of **3**.

6) Are the intermediates detected in ED Fig. 5b also found in the deletion mutant? Are the intermediates detected in ED Fig. 5b labeled by the deuterium-labeled amino acids as one would expect?

Response:

We did not carry out this analysis because the measured peak areas for the potential products (**Extended Data Fig. 5**) show that the yields are 1000–10,000-fold lower than those of virginiamycin M **1** (this yield calculation is based on the assumption that the other metabolites are related to **1**). As we are already at the limit of detection for incorporation of amino acids into compound **3** (100-fold lower yield than **1**), it was not possible to carry out analogous studies with these additional metabolites. Furthermore, their low yields meant that we couldn't reliably detect them even in repeat fermentations of the same strain, and thus it would not have been possible to confidently interpret their absence in the interface deletion mutant.

7) For ED Fig. 7, part c, identification of smaller fragments of **3** would help to verify its structure.

Response:

We thank the reviewer for this constructive comment. To this end, we have reacquired MS² data on metabolite **3**, under new chromatographic conditions that permitted a greater separation from metabolite **1** (it is clear now that the previous overlap between the **1** and **3** peaks led to the disappearance of certain fragments from the spectrum of **3** during background subtraction). These newly-obtained data confirm and extend our initial results by revealing previously-unobserved fragments whose structures we can rationalise (see revised **Extended Data Fig. 7**). Notably, three of the new **3** fragments are shared with both metabolites **1** and **2**, further confirming the relatedness of the three compounds.

8) In part d, the 260.0917 fragment does not seem reasonable (somehow the oxazole and the carboxylate are fused together with removal of proline?). In part d, it seems that ring opened forms of all of these product ions are more likely than ring closed forms.

Response:

The 260.0917 fragment arises by hydrolytic loss of (dehydro)proline to give the oxazole carboxylic acid (e.g. addition of H₂O). As the exact mass is the same, the structures can be drawn in either ring-opened or ring-closed forms.

9) For ED Fig. 8, metabolites **1** and **2**, part b, why is +1D not seen in the spectrum?

Response:

We thank the reviewer for raising this issue, and have now reassessed the data. As noted in our initial version of the ms (**Supplementary Fig. 6**, now renamed **Supplementary Fig. 7**), the resolution at which we carried out these experiments (60 K, 200 *m/z*, full width at half its half maximum (fwhm), which translates into a resolution of 40K at

500 m/z) doesn't allow distinction between two ions which have a $\Delta m/z = 0.0030$ AMU – as is the case with metabolites differing by the alternative presence of ^2H (deuterium) (+1.0063) and ^{13}C (+1.0033). At 40K, discriminating between such ions would have required a mass difference of 0.0125 AMU or greater (i.e. 4× the theoretical difference between ^2H and ^{13}C). To reflect this ambiguity, we have re-labelled the peaks in the spectra of both **Extended Data Fig. 8** and **Supplementary Fig. 7** as +1, +2, etc. to indicate that they potentially represent a mixture of isotopically-labelled species arising from natural abundance ^{13}C and incorporation of deuterium.

Nonetheless, the presence of the labelled amino acids in metabolites **1–3** is clearly demonstrated by two observations: i) the increase in intensity of the +1 peak in the presence of L-serine-2,3,3- D_3 (**Extended Data Fig. 8**), and the corresponding appearance of a +2 peak in the presence of L-proline-2,5,5- D_3 which is not observed in the absence of feeding (**Supplementary Fig. 7**); and, ii) the appearance of +2, +3 and +4 peaks when both L-serine-2,3,3- D_3 and L-proline-2,5,5- D_3 are fed.

Furthermore, based on the premise that each peak represents a mixture of isotopomers, we can rationalize the overall observed peak intensities. A sample calculation is as follows (**Extended Data Fig. 8c**, metabolite **1** (fermentation in the presence of L-serine-2,3,3- D_3 and L-proline-2,5,5- D_3):

Sum of all of the integrals = 298 (molecules)

Number of molecules (/298)	Isotopomer	Label	Comment
100	Unlabelled	[M] H^+	
32	$^{13}\text{C}_1$	+1	$^{13}\text{C}_1$ should be 32% of the base peak according to the unlabelled spectrum.
8	$^2\text{H}_1$ (i.e. 1D)		
2.6	$^2\text{H}_1$ (1D) + $^{13}\text{C}_1$	+2	Red from labelled serine Blue from labelled proline
87.4	$^2\text{H}_2$ (2D)		
28	$^2\text{H}_2$ (2D) + $^{13}\text{C}_1$	+3	Blue from labelled proline Green from proline and (serine or its metabolic product, glycine, which also introduces one ^2H into the molecule)
20	$^2\text{H}_3$ (3D)		
6.4	$^2\text{H}_3$ (3D) + $^{13}\text{C}_1$	+4	Green from proline and serine/glycine
10.6	$^2\text{H}_4$ (4D)		
3.4	$^2\text{H}_4$ (4D) + $^{13}\text{C}_1$	+5	Not included in the original data in order to simplify the presentation, but present (see spectrum below)

MS of metabolite **1**, showing the full range of isotopomers:

This calculation was carried out as follows:

1. The +1 peak must arise from either the ^{13}C isotopomer (no D) or incorporation of ^2H (i.e. 1D) from serine (or serine metabolized to glycine, which would also introduce a single D label). From the experiment in the absence of added label, the ^{13}C contribution is 32%, and therefore the remainder of the +1 peak (8%) must arise from ^2H .
2. The +2 peak must arise from incorporation of one ^2H + one ^{13}C , or from two ^2H (i.e. the two deuteriums of the labelled proline). As above, the +1 peak arising from $^2\text{H}_1$ incorporation will give rise to a +2 peak due to the additional presence of ^{13}C ($32\% \times 8\% = 2.6\%$). The remainder of the +2 peak (87.4%) is then attributable to incorporation of proline.
3. The +3 peak must represent a combination of $^2\text{H}_2$ (i.e. 2D) (from proline) + ^{13}C , and $^2\text{H}_3$ (i.e. incorporation of 3D from both proline and serine/glycine). ($^2\text{H}_2 + ^{13}\text{C}$) is present at 32% of the previous $^2\text{H}_2$ peak ($32\% \times 87.4\% = 28\%$), and thus the remainder (20%) must arise from incorporation of $^2\text{H}_3$.

And so on. Each $^{12}\text{C}_{28}$ peak must have an associated $^{12}\text{C}_{27}/^{13}\text{C}_1$ peak at +1 mass units and at 32% of the size, while the remainder of the isotopic peak arises from incorporation of ^2H (D) alone.

Overall, we can calculate the incorporation of serine/glycine and proline as follows:

51/298 molecules from serine/glycine = 17% incorporation

155/298 molecules from proline = 52% incorporation

Double incorporation should thus be $17\% \times 52\% = \text{ca. } 9\%$ (roughly as observed, 10.6%).

We have now amended the legends to both **Extended Data Fig. 8** and **Supplementary Fig. 7** to reflect all of these considerations.

10) For ED Fig. 8, metabolite 3, part c, why is +4D not seen in the spectrum?

Response:

Metabolite **3** was detected at 100-fold lower intensity than metabolites **1** and **2**. The incorporation peaks are at lower intensity still. Thus, in this case, while a +2D peak was clearly observed for incorporation of L-serine-2,3,3-D₃, and the intensity increased in the presence of L-proline-2,5,5-D₃ (by ca. 4%), any peak corresponding to simultaneous incorporation of the two labelled amino acids (already of low probability) was below the limits of detection.

11) The authors make this statement: "...while the presence of substrate analogues did not increase but moderately diminished affinity. The latter result, which implies substrate tolerance, ..." The logic behind this statement is unclear.

Response:

This phrase (lines 92–93), which was indeed confusing, has been removed.

Reviewer #3 (Remarks to the Author):

The authors provide insights into the mechanism underlying β -methylation in trans-AT PKS systems, thereby using vigninamycin biosynthesis as an example. By structural biology, they reveal the basis for substrate choice by recognizing the respective domains.

In general, the analyses performed are sound and the conclusions seem to be reasonable for the system used.

However, it might be of interest to test if this is a general theme applicable to other ACP β -methylation enzyme "pairs". If I got the authors right, the structure of the ACPs is quite conserved. Crystal structures exist, and if not alpha fold should be able to provide good predictions. For β -methylation enzymes, also structures were generated. Hence, would it be possible to model this on a bigger level? This analysis should be done to deduce (at least *in silico*) if the residues that were shown to be essential in complex formation are crucial in general. (Being aware that the authors wrote: "understanding the detailed role played by these residues in the interaction awaits higher resolution structural information".)

Response:

Please see response to reviewer 1, point 3).

1) Some points that might help to make the manuscript more clear to the reader:

Line 126: The residues mentioned here are not specifically depicted in Figure 2c. Would it be possible to do so (thereby, of course keeping the readability of the figure)?

Response:

We thank the reviewer for this comment. Unfortunately, given the structural view, it is not possible to indicate these residues in the figure. However, they have been highlighted in **Extended Data Fig. 4**.

2) Line 190: Even though in the context it becomes clear, the sentence is not directly clear which charge is disfavoring complex formation. In Figure 4, the label of the residues is hard to read (especially black on dark blue). It should be referred to a figure to clarify this point.

Response:

We did not mean to imply that a single charge disfavours complex formation. Rather it is the overall electrostatic surface potential of the domains – which differ between the investigated ACPs – that encourages or discourages complex formation. In any case, the question is not whether the interaction is absolutely forbidden, but rather that it must be slower than competing interactions (most typically *in cis*), and so doesn't occur to any measurable extent. To clarify this point, we have modified the text as follows:

Thus, both ACP_{5a} and ACP₆ exhibit positive net charge in regions which are negatively-charged in ACP_{5b} and ACP₇, aggregate electrostatic features which we propose disfavour productive complex formation with the β -cassette enzymes.

3) Line 234: Please elaborate on why “this modification is reduced under native biosynthetic conditions”.

Response:

To clarify this point, we have modified the phrase as follows:

Therefore, while Vir ACP₇ is recognised with good affinity by the β -methylation cassette, the low yield of 3 compared to 1 and 2 shows that this modification is reduced under native biosynthetic conditions (Fig. 1a).

4) Line 281: The speculation that *holo*-ACP_{5b} is not an efficient substrate for malonylation by the *trans*-acting AT could be easily tested using the methodology described in the manuscript. This should be experimentally be done. In addition, the complex formation between ACP, β -methylation cassette and *trans*-AT can be tested (maybe using binding affinities as done by the authors).

Response:

We would have liked to carry out this experiment, but despite extensive attempts to obtain the *S. virginiae* enzyme using *E. coli*, *Streptomyces coelicolor* and yeast as heterologous hosts, we were unsuccessful, as the protein was poorly expressed and/or completely insoluble. Below are representative gels which illustrate the results of expression in a.: *S. coelicolor*, and b. yeast, following cell lysis. The label 'I' refers to the insoluble fraction, and 'S' to the supernatant. The protein (expected MW 30.5 kDa) was poorly expressed in *S. coelicolor* and largely found in the insoluble fraction, while expression from yeast was even poorer.

KEY:

1. 30 mM Tris-HCl pH 8.5, 400 mM NaCl
2. 20 mM Tris-HCl pH 8.0, 400 mM NaCl, 10% glycerol, 5 mM EDTA
3. 20 mM Hepes pH 8.5, 400 mM NaCl
4. 50 mM NaPi pH 7.5, 250 mM NaCl, 10% glycerol

5) Line 305: The authors propose that deblocking latent chemistries would diversify polyketide structures. To me, it remains a bit unclear how that should be done. In the study, only minor side products had been identified. However, this was not triggered in a rational way.

Response:

The referee is correct in noting that we have not yet demonstrated deblocking. Nonetheless, the existence of an imperfect control mechanism in the system, and the fact that a large number of *trans*-AT PKS systems incorporate *trans*-acting enzymes which must compete with *in cis* modifications, suggests that these points of 'weakness' might be exploited to boost the yield of minor side products or even direct the systems towards the synthesis of novel metabolites. We are currently exploring this possibility experimentally, but only mean to propose the idea here.

6) Line 919: In the Extended Data Fig. 5 (and the others) the reads of the chromatograms are always given on a scale from 0 to 100. It would be interesting to see the absolute values to judge about the abundance of a signal.

Response:

We believe it is more informative to report calculated peak areas rather than absolute values (as in **Fig. 5, Extended Data Figs. 5 and 6**), as assuming that the compounds are related to virginiamycin M **1**, the areas can be directly converted into yield estimates via the **1** standard curve (**Supplementary Fig. 5**). In any case, the message of **Extended Data Fig. 5** is that the concentrations of the potential metabolites are so small as to be insignificant relative to **1-3**.

Reviewer #4 (Remarks to the Author):

Collin and coworkers report a fundamental study on the selectivity of beta-alkylation in virginiamycin M biosynthesis via the detailed characterization of protein structures and protein-protein interactions of VirC/D/E and ACPs. The authors demonstrate that ACP5b is the scaffold for beta-alkylation, and surprisingly, ACP7 also provides another possible scaffold. The authors further show the existence of a trace amount of the doubly beta-methylated product in the cells. In addition, the authors succeed in proposing some key mechanisms for the distinguishment of acyl-ACP substrates to be beta-alkylated. The manuscript is well written, the data are very solid, and the story is fascinating. This study is important for understanding the basal mechanisms of beta-alkylation of polyketides, and it will open the new way for modifying the chemical structures of polyketides by controlling beta-alkylation.

Unfortunately, however, some description and figures are not clear to understand. **Extended Data Figures are messy throughout.** This manuscript can be accepted after some revisions as below.

Response: The **Extended Data Figures** have been extensively revised to improve their appearance.

Major comments.

1) Introduction. It may be enough for the PKS scientists, but it is hard for general readers. I suggest the authors to include basic information, e.g., polyketides are... and the fact that beta-alkylation resembles the mevalonate pathway.

Response:

We thank the reviewer for this suggestion. In response, we have added the following sentences to the introduction:

The polyketide specialised metabolites of bacteria exhibit a diverse range of biological activities, including antibiotic and anti-cancer properties, and are heavily employed as drugs^{1,2}.

Relative to the *cis*-AT PKSs, the *trans*-AT systems⁵ (**Fig. 1a**) incorporate one or more free-standing enzyme activities and a wider variety of enzymatic functions, including cassettes of enzymes which introduce β -branching into the polyketide intermediates⁶. A common modification is β -methylation, whose chemistry is reminiscent of the mevalonate pathway of isoprenoid biosynthesis^{6,7}.

2) Introduction, Lines 52-67. This paragraph is redundant.

Response:

We believe that this paragraph is necessary, as it provides a brief summary of the major results of the study, which is standard for the journal.

3) In the manuscript, the authors use the term “VirC-E” as the meaning of “VirC, VirD, and/or VirE”. However, this confuses the readers because it also seems like “VirC-VirE complex”. Particularly, in Line 79, “binding of VirC-VirE to the holo-ACP5a-ACP5b” is so complicated.

Response:

We thank the reviewer for this useful comment. We have replaced VirC-E with VirC, VirD and VirE throughout.

4) Line 267. I felt a bit uncomfortable with “even when excised from its modular context”. Actually, ACPs were truncated from the original megaenzymes in this study. Then, do the authors think that the truncated ACPs have different conformations compared to those in the original megaenzymes?

Response:

The ACPs have indeed been ‘excised from their modular context’, as they have been expressed as discrete proteins. Nonetheless, their structural integrity is intact, as demonstrated by the solved NMR structures (**Extended Data Fig. 3**), which confirm that they all adopt the classical ACP 4 α -helix bundle folds.

5) Line 269. “The gate-keeping function within the cassette is therefore not limited to the HMGS VirC”. Difficult to understand the connection with the previous sentence.

Response:

We thank the referee for this comment. To clarify the intended meaning, we have revised the paragraph as follows:

We show here that β -methylation cassette members VirC, VirD and VirE do indeed preferentially recognise ACP_{5b}, even when the ACP is excised from its modular context, and that β -modification occurs within defined ACP_{5b}/partner complexes (**Fig. 2, Extended Data Fig. 3**). The fact that VirD and VirE prefer ACP_{5b} also demonstrates that the gate-keeping function within the cassette is not limited to the HMGS VirC.

Minor comments.

1) Fig. 1a. Modules 3, 8, and 10 are shown with gray bars. Do these show that these modules are NRPSs? Module 9 does not harbor its growing chain, so does this show that this module is not involved in chain elongation? These explanations should be included in the figure caption.

Response:

The referee was indeed correct that NRPS modules have been indicated with grey bars, and that module 9 is inactive for chain extension. Both of these elements are now explained in the accompanying figure legend.

2) Fig. 1b. Left structure is designated as VirAB. Is this correct?

Response:

Yes. We designated the ACP of the β -methylation cassette VirAB in previous work (doi: 10.1039/C3SC53511H), as it had not been identified during the original sequencing of the cluster (doi: 10.1016/j.gene.2006.12.035).

3) Fig. 4. Letters of amino acid residue numbers are not clear.

Response:

We have improved the clarity of this figure.

4) Extended Data Table 1. This table includes much important information. Nevertheless, it is a little bit bothersome to compare the results. My suggestion would be: ACPs are shown in the first column, then modifications (apo-, holo-, etc.) are shown in the subcolumn. The authors do not necessarily have to follow this, but can the authors reorganize the Table?

Response:

As suggested by the reviewer, we have added a column in the table indicating the form of the investigated ACP domain.

5) Extended Data Fig. 1a. Stacking gel can be cut. The index of MW markers is out of position.

Response:

We elected to show the unmodified gels (the second of which lacks the stacking gel). We thank the referee for noting the incorrect alignment of the MW markers, and this error has now been rectified, as well as the incomplete labelling of the lanes.

6) Extended Data Fig. 4. Suits are too small and hard to see. NCBI accession numbers for all proteins are required.

Response:

This figure has been extensively modified (see response to Referee 1, minor errors), and the NCBI accession numbers have been added, as requested (and included as well in new **Supplementary Figs. 8–13**).

7) Supplementary Figure 4. The R² value is 1.00, but generally it means that all points are on the line. In this case, it is due to the number of digits. Can the authors increase the digits to show that the value is not actually 1.00? Then, the caption should also be changed.

Response:

This modification has been made. The value is 0.997, and this figure has been corrected in the legend.

REVIEWERS' COMMENTS

Reviewer #1 (Remarks to the Author):

The revised manuscript has been extensively modified and my specific comments and comments shared by other reviewers have been addressed. The key issue of generalisation that was picked up on several occasions has now been satisfactorily addressed and helps set the manuscript firmly as one that will open up this field for further study.

Briefly then around two minor points.

2)Response:

We are unclear what the referee intended to say here concerning the raw data, as the SAXS curves are systematically presented.

I was referring to the fact that raw SAXS data plots for VirD and VirE (alone, not in complex with ACPs) was not shown. I meant that this would have been useful, given some of the challenges with the fits to complexes and to show a good fit could be obtained in these simpler cases, especially for VirD where the X-ray structure was available at high-resolution (granted this number is available in Supplementary Table 3 and the fit is good at 2.10). The authors have now expanded the extended figure 3 to include raw data for VirE (panel C). SAXS of the complexes is a significantly different challenge but one they have approached rigorously and underpins other studies in this area.

Extended data 3b) Is the theoretical red curve the OLIGOMER fitting? It is not stated in the legend but presume this is the case and this should be amended.

Under point 5) I think it is only necessary to add the Haines reference (10) alongside reference 15 as the addition of 'Further evidence that helix 3 plays a role in HMGS recognition of ACPA was provided by mutation of an ACPA of the mupirocin PKS, which resulted in a 3–10 fold decrease in mupirocin production in vivo (10)' breaks the flow of that paragraph.

Reviewer #2 (Remarks to the Author):

The authors have responded well to my criticisms. The only thing that I would suggest is that regarding whether *S. virginiae* makes 3, they not revise their manuscript so definitively. They detect a teeny tiny peak and have no MS/MS information. I would suggest rather than making the yellow edits, they simply add: "We were able to detect by LCMS a peak consistent with 3 in *S. virginiae*. However, given that it is produced in an extremely small amounts, we were unable to characterize it definitively." Or something to that effect.

Reviewer #3 (Remarks to the Author):

After reading the latest version of the manuscript as well as the reviewer comments and the answers of the authors, I believe the manuscript can be published.

The point raised by several reviewers if the findings can be generalized was now addressed by further sequence analyses (the key specificity elements were more likely to be conserved).

Overall, the revised version is improved and it was a pleasure reading the manuscript.

Reviewer #4 (Remarks to the Author):

My Major comment (2) might be misleading.
It was that, the paragraph is a bit too long and it can be shortened.

Other than this, the authors have addressed all my concerns.
I look forward to seeing this excellent work published.

RESPONSE TO REVIEWERS

REVIEWERS' COMMENTS

Reviewer #1 (Remarks to the Author):

The revised manuscript has been extensively modified and my specific comments and comments shared by other reviewers have been addressed. The key issue of generalisation that was picked up on several occasions has now been satisfactorily addressed and helps set the manuscript firmly as one that will open up this field for further study.

Response

We thank the referee for this positive assessment.

Briefly then around two minor points.

2) Response:

We are unclear what the referee intended to say here concerning the raw data, as the SAXS curves are systematically presented.

I was referring to the fact that raw SAXS data plots for VirD and VirE (alone, not in complex with ACPs) was not shown. I meant that this would have been useful, given some of the challenges with the fits to complexes and to show a good fit could be obtained in these simpler cases, especially for VirD where the X-ray structure was available at high-resolution (granted this number is available in Supplementary Table 3 and the fit is good at 2.10). The authors have now expanded the extended figure 3 to include raw data for VirE (panel C). SAXS of the complexes is a significantly different challenge but one they have approached rigorously and underpins other studies in this area.

Response

In response to the reviewer's comment, we have now added SAXS data obtained on isolated VirD (**Supplementary Fig. 6b**).

Extended data 3b) Is the theoretical red curve the OLIGOMER fitting? It is not stated in the legend but presume this is the case and this should be amended.

Response

The referee is correct. We have now added '(in red)' to the legend (now **Supplementary Fig. 6c**) to indicate the OLIGOMER fitting.

Under point 5) I think it is only necessary to add the Haines reference (10) alongside reference 15 as the addition of 'Further evidence that helix 3 plays a role in HMGS recognition of ACPA was provided by mutation of an ACPA of the mupirocin PKS, which resulted in a 3–10 fold decrease in mupirocin production in vivo (10)' breaks the flow of that paragraph.

Response

As suggested by the referee, we have removed the sentence, and added the suggested Haines reference at this point.

Reviewer #2 (Remarks to the Author):

The authors have responded well to my criticisms. The only thing that I would suggest is that regarding whether *S. virginiae* makes 3, they not revise their manuscript so definitively. They detect a teeny tiny peak and have no MS/MS information. I would suggest rather than making the yellow edits, they simply add: "We were able to detect by LCMS a peak consistent with 3 in *S. virginiae*. However, given that it is produced in an extremely small amount, we were unable to characterize it definitively." Or something to that effect.

Response

In fact, we do have MS/MS data to support the assignment of metabolite 3 in *S. virginiae* extracts. We have added these data to the figure (now **Supplementary Fig. 12**). In light of this, we would prefer to keep the revised text in the manuscript.

Reviewer #3 (Remarks to the Author):

After reading the latest version of the manuscript as well as the reviewer comments and the answers of the authors, I believe the manuscript can be published. The point raised by several reviewers if the findings can be generalized was now addressed by further sequence analyses (the key specificity elements were more likely to be conserved). Overall, the revised version is improved and it was a pleasure reading the manuscript.

Response

No changes required. We appreciate the referee's positive remarks.

Reviewer #4 (Remarks to the Author):

My Major comment (2) might be misleading. It was that, the paragraph is a bit too long and it can be shortened.

Response

In the absence of any evident way to shorten the paragraph while communicating the full set of key results, we would prefer to maintain it as is.

Other than this, the authors have addressed all my concerns. I look forward to seeing this excellent work published.

Response

We appreciate the referee's encouraging assessment.

REVIEWER COMMENTS

Reviewer #1 (Remarks to the Author)

The authors have addressed the final minor points.